# CryoEM reveals the stochastic nature of individual ATP binding events in a group II chaperonin

Yanyan Zhao [1], Michael F. Schmid [2], Judith Frydman [1,3,4,5] & Wah Chiu [1,2,6✉]

Chaperonins are homo- or hetero-oligomeric complexes that use ATP binding and hydrolysis to facilitate protein folding. ATP hydrolysis exhibits both positive and negative cooperativity. The mechanism by which chaperonins coordinate ATP utilization in their multiple subunits remains unclear. Here we use cryoEM to study ATP binding in the homo-oligomeric archaeal chaperonin from *Methanococcus maripaludis* (MmCpn), consisting of two stacked rings composed of eight identical subunits each. Using a series of image classification steps, we obtained different structural snapshots of individual chaperonins undergoing the nucleotide binding process. We identified nucleotide-bound and free states of individual subunits in each chaperonin, allowing us to determine the ATP occupancy state of each MmCpn particle. We observe distinctive tertiary and quaternary structures reflecting variations in nucleotide occupancy and subunit conformations in each chaperonin complex. Detailed analysis of the nucleotide distribution in each MmCpn complex indicates that individual ATP binding events occur in a statistically random manner for MmCpn, both within and across the rings. Our findings illustrate the power of cryoEM to characterize a biochemical property of multi-subunit ligand binding cooperativity at the individual particle level.

[1] Biophysics Graduate Program, Stanford University, Stanford, CA, USA. [2] Division of CryoEM and Bioimaging, SSRL, SLAC National Accelerator Laboratory, Menlo Park, CA, USA. [3] Department of Biology, Stanford University, Stanford, CA, USA. [4] Department of Genetics, Stanford University, Stanford, CA, USA. [5] CZ Biohub, San Francisco, CA, USA. [6] Department of Bioengineering, James Clark Center, Stanford University, Stanford, CA, USA. ✉email: wahc@stanford.edu

Proteins must fold into their correct three-dimensional conformations in order to function properly. Nonnative protein conformations can lead to aggregation, and results in loss-of-function, dominant-negative effects or even toxic cellular effects[1]. Aberrant folding has been linked to a rapidly expanding list of pathologies, including Alzheimer's disease (AD) and Parkinson's disease (PD), as well as Type II diabetes and certain forms of heart disease and cancer[2]. The information required for a protein to adopt the correct tertiary structure resides in its amino acid sequence. For a very small set of proteins, folding appears not to require additional cellular factors. However, the vast majority of cellular proteins are unable to reach their native state spontaneously[3]. The correct folding of these proteins requires the assistance of a complex cellular machinery of molecular chaperones[4].

Chaperones comprise several protein families involved in various processes including folding, unfolding, and homeostasis of cellular proteins[5]. Chaperonins are a highly conserved subgroup of chaperones that consists of two stacked seven- to nine-membered rings. Chaperonins bind nonnative polypeptides in their central cavity and consume ATP to mediate correct polypeptide folding[6]. Chaperonins are further divided into two major subtypes, group I and group II. Both share a similar domain arrangement, in that each subunit consists of three domains: an equatorial ATP-binding domain; an apical domain that is involved in substrate binding; and an intermediate domain that enables communication between the equatorial and apical domains. Group I chaperonins, mainly found in eubacteria, mitochondria and chloroplasts[7], such as GroEL of E. coli, function in conjunction with a ring-shaped cofactor, such as GroES, that caps the cavity for polypeptide substrate refolding[5,8]. By contrast, group II chaperonins occur in archaea and eukaryotic cytosol and have a built-in lid segment protruding from the apical domain that opens and closes with ATP binding or hydrolysis[9].

The action of chaperonins is coupled to ATP driven conformational changes that are coordinated in time and space by complex allosteric regulation[10]. In the past decades, progress has been made in describing their various functional allosteric states, the pathways by which they interconvert, and the coupling between allosteric transitions and protein folding reactions[11]. Kinetic studies have shown that group I and group II chaperonins exhibit positive intra-ring and negative inter-ring cooperativity for ATP hydrolysis[12–14]. Intra-ring conformational changes in group I chaperonin GroEL are proposed to follow a concerted, Monod–Wyman–Changeux (MWC) model[15–17]. Each ring is in equilibrium between two states that interconvert in a concerted manner: a T state, with low affinity for ATP and high affinity for nonfolded protein substrates, and an R state with high affinity for ATP and low affinity for nonfolded protein substrates[18]. Molecular dynamics simulations have shown that steric repulsions between subunits prevent conformational changes in individual subunits unless the intra-ring transition is concerted[19,20]. In contrast to GroEL, structures of the open conformation of group II chaperonins from archaea reveal few lateral contacts between the intermediate or apical domains of neighboring subunits, indicating there are no steric impediments which would prevent independent conformational changes in individual subunits[9,21–23]. The regulation of eukaryotic chaperonin TRiC/CCT is likely more complex. First, it has eight distinct subunits with different ATP binding affinities[24]. Second, close contacts between the apical domains of some of the neighboring subunits[25,26], suggest a possibly different intra-ring allosteric mechanism. The Koshland–Némethy–Filmer-type intra-ring allostery has been proposed for the hetero-oligomeric TRiC/CCT ATP binding and/or hydrolysis[27–29]. Whether the intra-ring cooperativity of group II chaperonins begins at the ATP binding step has not been investigated.

Negative inter-ring cooperativity in ATP binding/hydrolysis appears to be a universally conserved property of all chaperonins. The group I chaperonin GroEL is thought to have negative cooperativity, whereby the double-ring undergoes sequential ATP-induced Koshland–Némethy–Filmer (KNF) type transitions from the TT state via the TR state to the RR state[30]. However, the mode of inter-ring communication in group II chaperonins could be different, due to a different inter-ring subunit arrangement, namely staggered in group I versus in-register in group II chaperonins[31].

The MWC and KNF models employed to describe allostery in chaperonins suggested by ensemble biochemical assays do not distinguish allostery in ATP binding from that in ATP hydrolysis due to the fact that hydrolysis, which is easy to measure, is often assumed to be a proxy for binding. It is important to assess if these allosteric modes exist and how they relate to cooperativity. Recent observations of partial nucleotide occupancy in the open state structure of CCT[25] also raise the question whether all the ATP binding sites in one ring are sequentially (consecutively or not) filled and whether the intra-ring or inter-ring subunit conformations are influenced by ATP occupancy and distribution. These questions will illuminate the contribution of ATP binding vis-a-vis hydrolysis to the allosteric regulation in group II chaperonins. The availability of high-resolution structures representing different conformational states in solution will provide molecular details of the conformational cycle.

Previous studies examining nucleotide binding in chaperonins measured the distribution of different ATP occupancies in chaperonin complexes using native mass spectrometry[17] and single molecule fluorescence microscopy[32]. While these techniques can track the number of nucleotides bound to chaperonin molecules, they do not address questions about allosteric control of ATP binding events, e.g., whether they occur in adjacent or distal subunits in each chaperonin. To determine whether ATP binding to group II chaperonins follows a cooperative behavior and if so, whether it follows a specific allosteric mode (MWC or KNF) requires approaches that would retain the positional information of ATP binding events within the double ring particle. To address this question, we developed a tiered approach that uses cryoEM single particle analysis to track both the number and the distribution of nucleotides bound to each subunit in the homo-oligomeric MmCpn archaeal chaperonin complex. Our approach allowed us to determine whether binding to one subunit enhances the likelihood of binding to an adjacent, as opposed to a more distal, subunit in the same particle. Examining this nucleotide distribution in each MmCpn subunit for hundreds of thousands of particles reveals that ATP binds to the chaperonin MmCpn in a statistically random manner both within a ring and across the rings. This random binding mode leads to conformational and compositional heterogeneity of intra-ring subunits. These analyses indicate there is no cooperativity in ATP *binding* to archaeal group II chaperonins at the ATP concentrations we study.

## Results

The aim of our investigation is to use cryoEM to determine the binding occurrence of ATP to the subunits at the individual particle level. We develop the following image processing strategy to answer this biochemical question and thus do not necessarily aim to derive an ensemble averaged structure of MmCpn as cryoEM structure determination normally does.

**Initial cryoEM map of MmCpn to be used for identifying the subunit positions needed for subsequent subunit structure classification.** To assess the mode of ATP binding by MmCpn, we incubated the chaperonin with a low, non-saturating

concentration of ATP and froze immediately. In brief, MmCpn was purified as described in the methods[33]. To demonstrate that the MmCpn preparation used here is fully active and functional, we performed three orthogonal assays well accepted in the field of chaperonin biochemistry: a proteinase K (PK) digestion assay, a rhodanese folding assay, and steady state ATPase measurements at a range of ATP concentrations (Supplementary Fig. 1). All three assays confirmed MmCpn can fully cycle to the closed ATP-hydrolysis induced conformation, can fold substrates and exhibit the expected allosteric hydrolysis regulation. These measurements agree with previous studies[13,33,34], indicating that the MmCpn preparation used is fully active and functional. MmCpn was then mixed with ATP at a molar ratio of 1 Cpn hexadecamer to 8 ATP molecules and the sample was frozen immediately on glow discharged grids to stop ATP hydrolysis. To avoid the over-representation of the images with two preferred orientations in the final reconstruction, the number of particles of these orientations are reduced to equalize the contributions of different particles' orientations. Around 160,000 particles with good Euler angle coverage were used for an initial map reconstruction with D8 symmetry (Fig. 1a, Supplementary Fig. 2b). Although the reconstructed map has an average resolution of 3.9 Å by FSC gold standard (Supplementary Fig. 2c), the local resolution varies from 3.5 Å at the equatorial domain to more than 10 Å at the apical domain (Fig. 1b). After map sharpening, we observe well-resolved side chains in the equatorial domain, but low average density in the intermediate and apical domains. The high resolution obtained for the equatorial domain by imposing the D8 symmetry in the reconstruction supports the fact that the equatorial domains in the complex are relatively invariant in response to ATP binding, and thus follow strict D8 symmetry. In contrast, the poor resolvability at the upper domain consisting of intermediate and the apical domains indicates the absence of D8 symmetry in these regions within or among the MmCpn molecules, i.e., the apical domains exhibit high conformational heterogeneity in response to ATP binding. Reconstruction with no symmetry (i.e.,

C1 symmetry) from the same dataset of particles generated the same result as with D8 symmetry reconstruction (Supplementary Fig. 3). These results demonstrate that, upon ATP binding, the MmCpn complex demonstrates low structural variability in the equatorial domains and high structural heterogeneity and flexibility in the upper apical domains, leading to the apical domains averaging out *among* the MmCpn particles (Supplementary Fig. 3). Therefore, using C1 symmetry or D8 symmetry leads to a similar initial model of MmCpn bound to ATP, where the equatorial domains are well resolved but the conformation of the apical domains, which presumably undergo the largest conformational change upon ATP binding, cannot be resolved. The following steps are to determine the structure of each subunit in each of this ensemble of particles.

**A cryoEM analysis pipeline to analyze ATP-induced heterogeneous conformational changes in the homo-oligomeric chaperonin MmCpn.** CryoEM is a structural tool potentially capable of unraveling the allosteric property of ATP binding to MmCpn but this will require determining the nucleotide bound subunit distribution within each chaperonin particle at different ATP occupancy levels. The group II chaperonin from Methanoccocus maripaludis, MmCpn is homo-oligomeric, consisting of two 8-membered rings stacked back-to-back. While the fully closed state induced by ATP hydrolysis is eight-fold symmetrical, the ATP-bound states can have different and heterogeneous conformations, depending on whether ATP occupies that particular subunit. Indeed, our initial analysis indicated there is high conformational variability in the apical domains but uniform conformation in the equatorial domains. This can be rationalized because, at each level of occupancy, there are distinguishable different arrangements of ATP-bound and ATP-free conformations in any given particle. Given all the possible rotational permutations of ATP occupancy in both the number of ATP-bound subunits per particle and the different ATP bound subunit distributions among particles, the chance to have two particles with the same apical domain conformation at the subunit level is extremely low. This may explain why the apical domains are not resolved if we use either D8 or C1 symmetry to obtain an averaged map which assumes each particle has the same conformation at the stated resolution.

To address this challenge, we developed a structural analysis pipeline consisting of three stages, as described in detail below. Briefly, the first stage exploited the fact that the equatorial domains are structurally invariant out to ~3.5 Å resolution to determine the orientations of the particles accurately using D8 symmetry (Fig. 1). This particle orientation information sets the stage for the second stage of analysis to extract individual subunits retaining their positional information in their holo-particles of origin. We then carried out focused classification over one million individual subunits using RELION without changing the subunit orientation. This allowed us to classify the apical domain conformations into unliganded and ATP-bound states (Figs. 2, 3). The number of subunits in each class was large enough to observe the individual subunit conformation and nucleotide occupancy at reasonable resolution (4–6 Å, Supplementary Table 2). Finally, we used the positional information that defines where these individual apical domain conformations come from in each one of the MmCpn particles. These allow us to determine the exact location of each subunit, either with or without ATP bound, in each individual particle, as well as the number of each subunit conformational states in each 8-subunit ring and 16 subunit complex. This pipeline allowed us to characterize statistically at a structural level, the ATP-binding in individual MmCpn and to determine the presence or absence of allostery in this process.

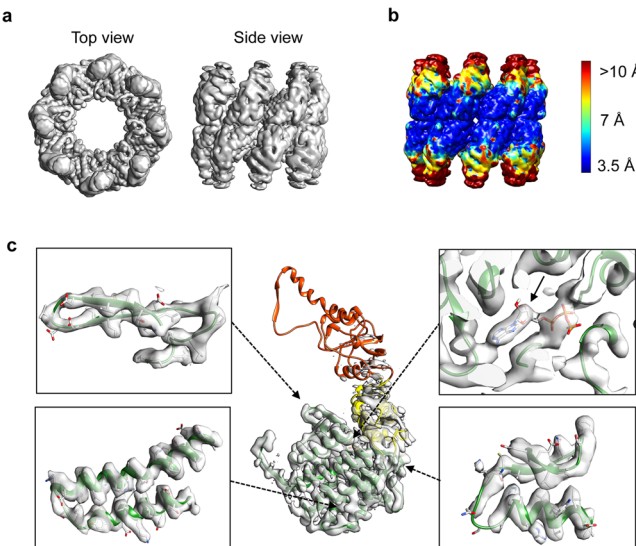

**Fig. 1 A 3.9 Å D8 symmetry averaged cryoEM structure of MmCpn in the presence of ATP. a** Top and side views of MmCpn cryoEM reconstruction. **b** Local resolution map calculated by MonoRes[51]. Color key for local resolution (in Å) is shown. **c** Segmented cryoEM map for a single subunit with the model superimposed. Equatorial, intermediate, and apical domains of the model are colored in green, yellow, and red respectively. High-resolution features in the equatorial domain are shown in side panels, including the ATP binding site (upper right with an arrow).

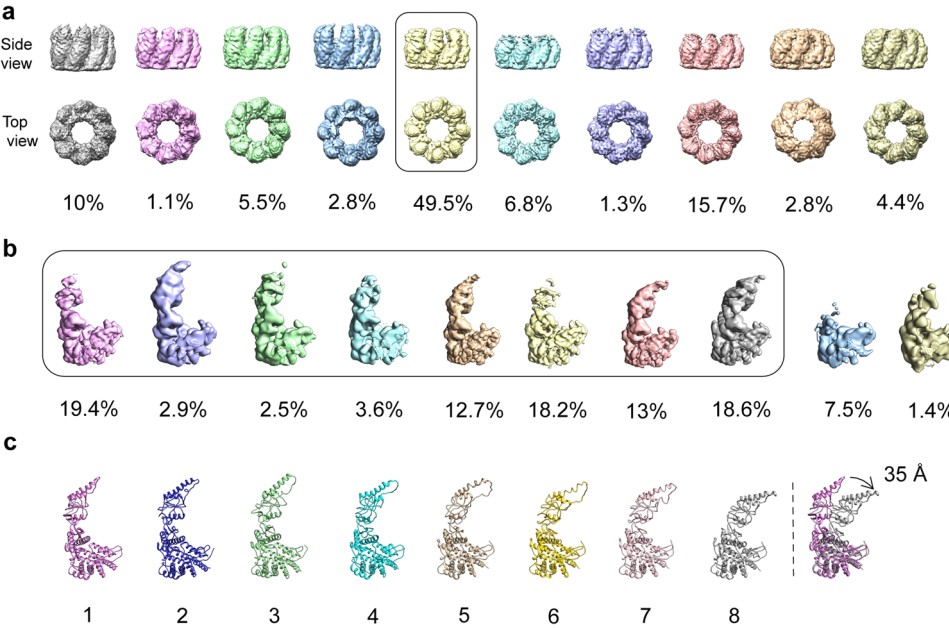

**Fig. 2 Focused classification of conformational heterogeneity. a** Single ring-focused classification into ten classes. The ring class outlined in black is selected for further analysis in (**b**). **b** Subunit-focused classification into ten classes of individual subunits. The subunit classes outlined in a black box are selected for modeling in (**c**). **c** Fitted models for corresponding subunit maps shown in (**b**). The distal tip of the apical domain moves inwards ~35 Å between the most open conformation (class 1) and the most closed conformation (class 8) (shown in the far right model with class 1 and 8).

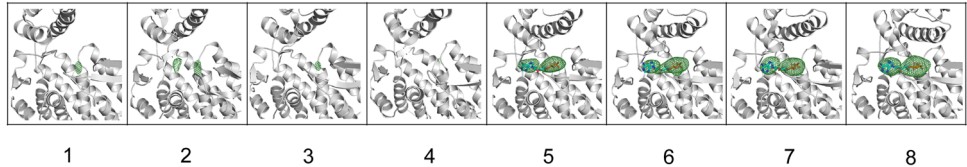

**Fig. 3 Difference map analysis of the nucleotide occupancy in each of the eight conformations.** Model for each conformation is generated by rigid body fitting and followed by real space refinement. The panels show the nucleotide binding pocket of each conformation. The density (in green mesh) represents difference density attributable to nucleotide in the binding pocket. A model of ADP is shown here to illustrate the orientation of the nucleotide in the binding pocket. The difference map is displayed at $+4\sigma$.

**Structural analysis of the conformation of each ring in each particle.** We chose the D8 symmetry reconstruction as the starting point for our subsequent steps of analysis as it allows a more accurate alignment at the equatorial domain among all particles to facilitate the comparison of the upper domain movement among several conformations. After fitting the model to the density of a single subunit, our analysis allowed us to attribute density corresponding to the nucleotide in the equatorial domain (Fig. 1c). Together, both the local resolution map and B-factor sharpened map demonstrate well-resolved density and connectivity at the equatorial domain, but large variability and poor density connectivity in the intermediate and the apical domains. To evaluate the compositional and conformational heterogeneity in our data, we first studied the conformation of each ring in each particle image. To do that, we performed focused classification[35] on a single ring of each particle as a particle unit. In brief, we generate a duplicate set of particles with Euler angle assignments that are equivalent to flipping the rings, using symmetry expansion option in RELION[35]. A soft mask is applied to the volume of a single ring in a 3-D model and then its projection images are computed according to the Euler angles. The 3D classification is performed by comparing the original particle images iteratively with the updated computed projection images yielding ten classes with distinct conformations (Supplementary Fig. 4). The final maps of these ten classes display different levels of ring closure (Fig. 2a) reflecting high heterogeneity

in ring conformations among particles. However, the subunits within the ring each adopt different conformations and compositions (i.e., with or without ATP) since we observe poorly resolved density at the apical domain in contrast to the equatorial domain for all the ring classes. By careful examination of the maps for each class, we chose to focus our analysis on one class accounting for around 50% of all the single ring particles, because the 3D map for this class appears the best resolved.

**Structural and conformational analysis of individual subunits in each ring.** To further understand the conformational heterogeneity at subunit level, we performed focused classification on each of eight subunits among the ~50% rings from the previous step (Fig. 2a). In brief, we generate another seven sets of particles with different Euler angle assignments equivalent to rotating 45° each time around the central axis of C8 symmetry to each symmetry point. We use the same computational procedure as described above to extract and classify the individual single subunit density in each ring. This operation yielded ten classes of distinct subunit conformations. Among the ten classes from 3D classification (Fig. 2b), two classes, accounting for less than 10% of all the subunit particles, were excluded in the next step analysis, as the subunit maps for these two classes are poorly resolved. For the remaining eight well resolved subunit classes, an MmCpn model (PDB ID: 3KFB) is rigidly

fitted into each subunit map and then refined by real space refinement from Phenix[36]. By comparison with published structures of open and closed form of MmCpn with imposed D8 symmetry[9,37], we find that these eight well resolved subunit classes covered conformations from the very open all the way to the very closed form (Fig. 2b, c, Supplementary Fig. 5). The upper domain consisting of the intermediate and the apical domain moves toward the equatorial domain, with maximum 35 Å displacement at the apical domain between the most open and closed conformations (Fig. 2c). The large space movement at the upper domain among subunits further explains the poor resolvability in this region of the full map reconstruction shown in Fig. 1.

**Nucleotide occupancy in each subunit**. To demonstrate the nucleotide occupancy in each subunit class, we calculated difference maps between the subunit density map for each class and the map calculated from the fitted MmCpn model of a single subunit. In the difference maps, we find that nucleotide density exists in the four more-closed conformations (class 5–8) but is absent in the four more-open conformations (class 1–4) (Fig. 3). This observation agrees well with previous studies that the nucleotide binding and/or hydrolysis can cause subunit conformational change from the open to the closed state[38]. Contrary to the previous studies, our analysis shows a variable extent of lid closure when ATP is bound at the subunit.

One technical concern in the above data processing protocol is that a significant amount of subunits in more-open conformations could be misclassified to the more-closed conformation groups and vice versa, since the masked classification method suffers from the potentially inaccurate extraction of targeted region of interest in the particle by comparing 2D particle image consisting of 16 subunits with projection image from a masked reference of a single subunit[35]. To validate our finding, we undertook an alternative data processing scheme to computationally extract a single subunit conformation (Supplementary Fig. 6a). In this approach, a mask to the 15 subunits other than the targeted subunit was applied to the D8 symmetry 3-D map and the corresponding projection image was computed for subtraction from the original particle image to generate a "synthetic" image with signal corresponding to only a targeted single subunit as done previously[39,40]. These "synthetic" images were subjected to iterative 3D classification with a single subunit mask. The resulting classes yielded similar results as classification without partial signal subtraction shown in Fig. 2b. That is, the closed conformation is not observed from reclassification of open form subunits, and vice versa (Supplementary Fig. 6b, d). The nucleotide occupancy for each conformation from reclassification also agrees with classification without partial signal subtraction, too (Supplementary Fig. 6c, e).

**Reconstructing intra-ring nucleotide occupancy and distribution at the individual particle level**. The next step is an entirely unique part of this analysis. For each ring of each particle, we mapped the two kinds of subunits, either apo- or ATP-bound, back into each of the 8 subunit positions around the ring from which they came, according to the 3D class they belonged to. We observe different levels of occupancy in the rings. In general, among the 50% of the rings analyzed, the majority have 4 to 6 nucleotide binding pockets occupied by nucleotide (Fig. 4a), consistent with the molar ratio of ATP to MmCpn subunits we used (one ATP for every two MmCpn subunits). However, various subunit-associated nucleotide distributions around the ring are observed for each nucleotide occupancy level. In an attempt to answer the question if the ATP binding to each MmCpn complex

is in a random or cooperative mode (e.g., whether adjacent subunits preferentially bind ATP), we predicted the relative frequency of each of the different possible arrangements of nucleotides per ring at each nucleotide occupancy level based on simple probability statistics estimation. For instance, if there are two ATPs in a ring, there are 4 probabilistic modes of their relative distributions in an 8-subunit ring if the ATP is bound randomly with no preference (Fig. 4b). Note that, for this two-ATP bound case (upper left set in Fig. 4b), for example, the adjacent-ATP pair (the leftmost of these four) can occur 8 different ways around the ring, as can the next two kinds of pairs, whereas the opposite-ATP pair (the fourth kind of pair) can only occur in 4 different ways, giving it half the predicted frequency as the others. A similar logic applies to all the cases. By comparing experimental observations of the relative frequency distribution of nucleotide occupancy at a subunit level within a ring with that based on a statistical prediction of random binding, we find that they agree well with each other regardless of the numbers of ATP per ring (Fig. 4b). This agreement indicates that ATP binding is a stochastic process and is not regulated via inter-subunit interaction within the MmCpn ring.

A commonly asked biochemical question regarding the cooperativity between subunits in an oligomeric macromolecule upon a ligand binding is if the conformational change in a subunit that is associated with ligand binding could influence the conformation of neighboring subunits which have not yet bound ligand. To examine this question in MmCpn binding to ATP, we consider the triplets of subunits with the middle subunit in the most closed form (class 8), while the left and right neighboring subunits in the open state (open classes 1,2,3,4, without ATP). The model is fit into the reconstructed triple-subunit map, and the superimposition of this model to the fully closed form cryoEM density map shows that the middle subunit can reach the closed form but has little effect on the openness or closedness of its nucleotide-unbound neighbors (Fig. 4c). In other words, our data shows no apparent allosteric effect on lid closure due to the neighboring subunit within the ring. To further support this point, we also performed similar analysis on triplets of subunits with the middle subunit in the most open form (class 1), while the left and right neighboring subunits in the closed state (closed classes 5,6,7,8, with ATP). The results are shown in the Supplementary Fig. 7.

**Reconstructing inter-ring nucleotide occupancy and distribution**. To examine the possible inter-ring allosteric regulation by ATP binding in MmCpn, we compare the nucleotide occupancy and distribution in both rings of an MmCpn molecule. ~80% of rings we are analyzing so far can be paired to their original double-ring MmCpn particle. By examining the nucleotide occupancy in these chaperonin particles, we find that most particles have similar nucleotide occupancy in both rings (Fig. 5a). This suggests that, unlike group I chaperonin GroEL which is negatively regulated through inter-ring ATP binding in the ensemble average measurements[41–44], ATP binding to the two rings of group II chaperonin MmCpn occurs in a stochastic manner. We next investigated the correlation between ATP binding in two subunits directly across the ring equator from each other. We calculated the relative frequencies of zero, one and two nucleotides bound to an inter-ring subunit pair at each ATP occupancy (Fig. 5b). As expected, we observe that the relative frequency for double occupancy gradually increases and eventually surpasses the single occupancy as ATP occupancy in the MmCpn complex increases. Then we calculated the difference between our observation and the random distribution at each fractional saturation (Supplementary Fig. 8). We find that our

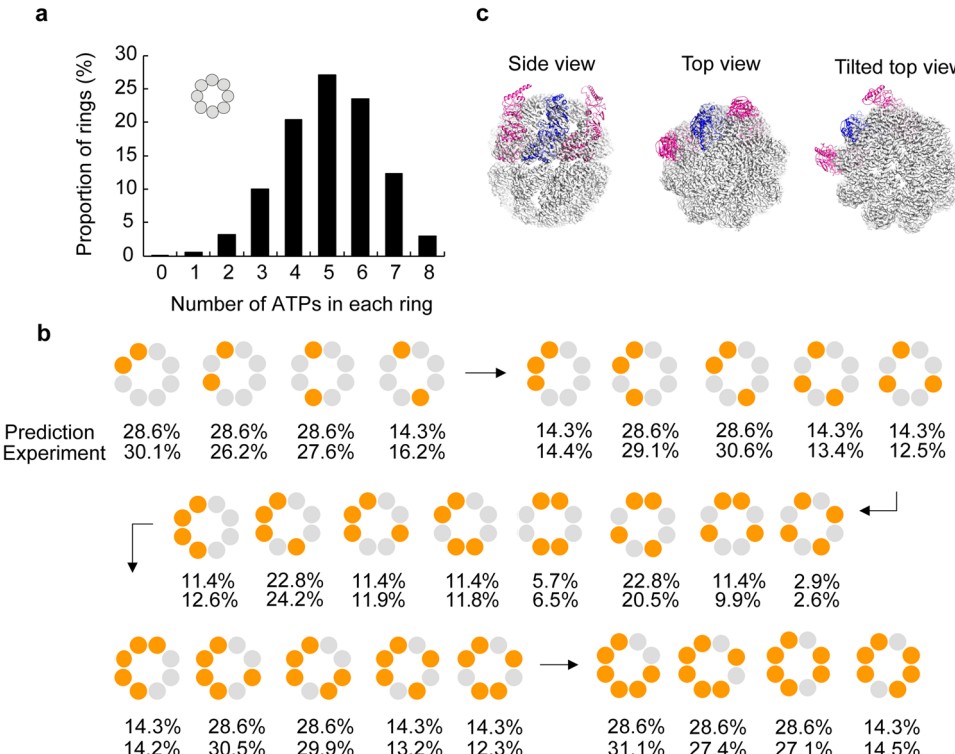

**Fig. 4 Intra-ring ATP occupancy and distribution. a** Nucleotide occupancy in the rings analyzed. **b** Probabilistic prediction and experimental observation of nucleotide distribution at different nucleotide occupancy levels (i.e., 2,3,4,5,6 nucleotides annotated as orange balls). **c** Model of the triple subunits superimposed on fully closed form (gray density) shows the middle subunit in the triplet in closed form while left and right subunits can exist in an open conformation.

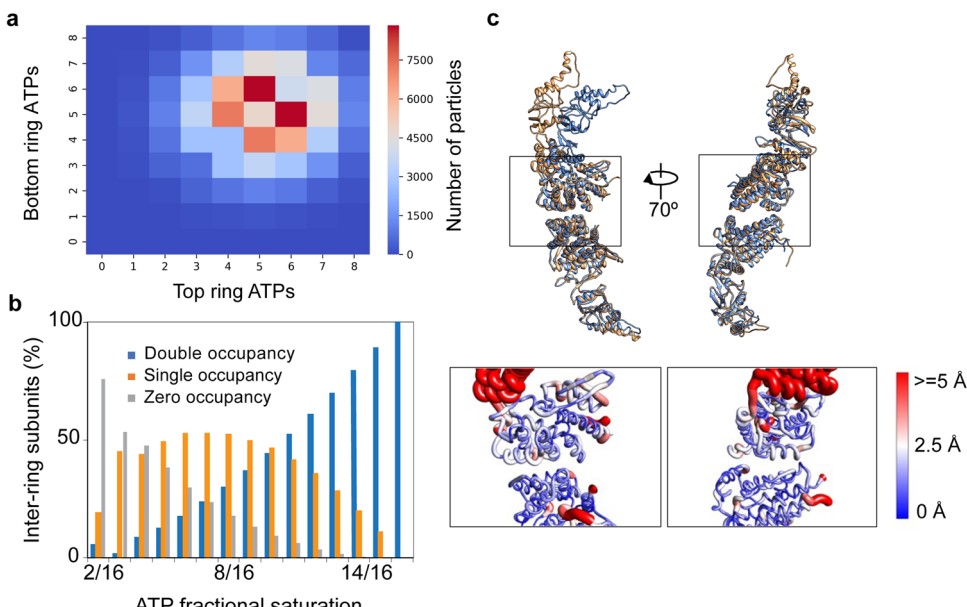

**Fig. 5 Inter-ring nucleotide occupancy and distribution. a** Nucleotide occupancy in ~80% of rings analyzed. **b** Nucleotide distribution in two subunits directly across the ring equator from each other at different ATP saturations. **c** Comparison of inter-ring interface between open/closed form (yellow) and closed/closed form (blue). Top panel shows the superimposition of the inter-ring subunit pair in open/closed form (yellow) and subunit pair in closed/closed form (blue). Bottom panel is the zoom-in view of the top panel but focuses on the inter-ring interface. The color bar and worm width represent the local backbone resolution variation level between the two models in the top panel.

observation is quite close to the random distribution prediction, especially at the middle to high fractional saturation range. This analysis indicates that the ATP binding in the inter-ring subunits is close to random binding.

Based on these two observations, we hypothesize that ATP binding introduces little change across the inter-ring interaction interface. To further test this hypothesis, we reconstructed a map of inter-ring subunit pairs with one subunit in nucleotide-bound

closed state (class 8) and the other subunit in nucleotide-unbound open state (class 1), or both subunits in the nucleotide-bound closed state (class 8). Models of both subunits were then rigidly fit into each map and refined using phenix.real_space_refine[45] (Supplementary Table 5). We then superimposed these two models to compare their inter-ring interaction interfaces (Fig. 5c). We observed little backbone variation at the inter-ring interaction interface as the subunit changed from the open nucleotide-unbound state (class 1) to the nucleotide-bound closed state (class 8). Taken together, these results indicate there is no negative inter-ring cooperativity for ATP binding, but rather random ATP binding to subunits across the ring interface. Furthermore, the relative independence of conformational change between individual rings suggests the existence of high heterogeneity across the rings.

**Analysis of intra-ring subunit conformational heterogeneity and allostery.** To understand the subunit conformational heterogeneity within each ring, we calculate the relative frequency of the eight different conformations at different nucleotide occupancy levels (Fig. 6a). We observe large conformational variation among eight subunits within the ring at each nucleotide occupancy level. Each subunit conformation can occur in the rings of different nucleotide occupancy (Supplementary Movie 1). This suggests that conformational change in chaperonin MmCpn is relatively independent in individual subunits of the ring. To investigate the occurrence of intra-ring interactions during the ATP induced conformational change, we map the footprints of subunit by superimposing eight models of different conformations (Fig. 6b). The subunit footprints indicate that the first intra-ring interaction is most likely to happen at the level of the helical protrusions of apical domains of subunits in the closed form. This further suggests that the observed cooperativity of ATP hydrolysis may not originate at the binding step, but likely at the ATP hydrolysis step. Unlike the homogeneous concerted model proposed for group I chaperonin GroEL, it appears that ATP binding in the group II chaperonin MmCpn leads to a heterogeneous non-concerted conformational change within the ring. The variation of intra-ring conformational heterogeneity among the ATP occupancy levels shown in Fig. 6a, together with the variation of ATP distribution at each ATP occupancy level shown in Fig. 4b, demonstrates the presence of the extraordinary heterogeneity in our study. To the best of our knowledge, it distinguishes our study from all previous cryoEM studies which pursue unique structures from subpopulations of tens of thousands to millions of identical particles and/or manipulation of the conditions to

purposely favor one or a few conformations. Our study thus demonstrates the power of cryoEM single particle analysis to study similarly extraordinary heterogeneous oligomeric complexes upon ligand binding in individual molecular components at an individual particle level.

## Discussion

Most previous studies of chaperonin allostery are based on ensemble averaged measurements. In particular, the nested positive and negative cooperativity observed for chaperonins including MmCpn via ATP hydrolysis measurements conflate the steps of ATP binding and hydrolysis. As the purpose of our study is to use cryoEM to determine the existence of allosteric modes (MWC or KNF) in ATP binding to oligomeric chaperonin MmCpn, the first step of this biochemical reaction at single molecule level, we analyzed the distribution of ATP bound subunits in individual MmCpn at various ATP occupancy levels. We conducted our study under a non-saturating ATP concentration where the most variability of the conformation and ATP occupancies of MmCpn at native state are captured. By freezing the sample immediately after addition of low concentration of ATP, we find that subunits from ~50% of the particles analyzed fall into two major groups, the nucleotide-free open conformation and nucleotide-bound closed conformation (Figs. 2, 3). This suggests that the majority of the ATP-bound subunits underwent the process of lid closure upon the gamma-phosphate recognition by the nucleotide sensing loop in the intermediate domain[46]. Since the subunit reopens the lid after ATP hydrolysis and phosphate release, some closed form subunits in our study may have ADP in the nucleotide binding pocket. Nonetheless, ATP binding stabilizes the closed conformations of the subunits. Interestingly, we capture some of the subunits appearing to be on the path of closure. Surprisingly, by tracking the nucleotide bound subunits in individual rings and particles, we find that ATP binding to the chaperonin MmCpn is stochastic both within the ring and across the ring. We also observe high-level subunit conformational heterogeneity within MmCpn molecules, indicative of the relative independence of subunit level conformational changes upon ATP binding. The lack of cooperativity between adjacent subunits and the heterogeneous non-concerted conformational change within the molecule demonstrates that neither the Koshland–Némethy–Filmer-type nor the Monod–Wyman–Changeux (MWC) allostery occurs during ATP binding at the ATP concentration we study.

As the energy currency in the cell, ATP is utilized by various types of proteins. Therefore, the local ATP concentration for

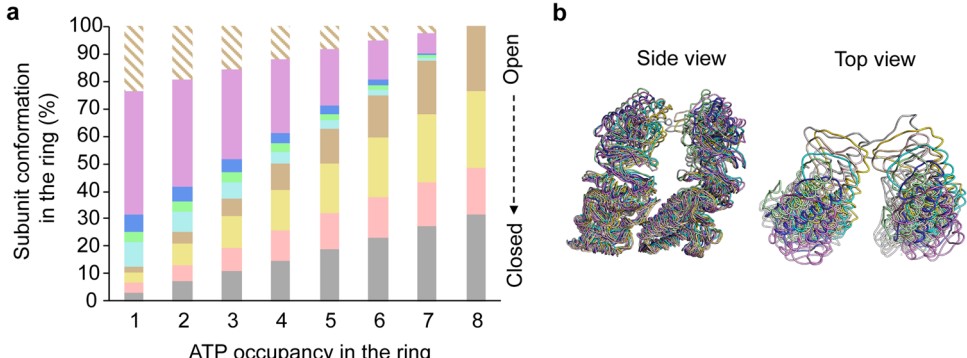

**Fig. 6 Intra-ring subunit conformational heterogeneity. a** The average intra-ring subunit conformation composition at different nucleotide occupancy levels. The colors are consistent with subunit maps and models of each conformation used in Fig. 2. For example, the gray color represents the most closed conformation (class 8) and the pink color represents the most open conformation (class 1) as shown in Fig. 2. The diagonal stripes represent the classes not analyzed because of poor map resolvability. **b** Side view and top view of intra-ring neighboring subunits in eight conformations. Model colors are consistent with the colors of histograms in a.

MmCpn in the cell is unknown. So far, we observe that ATP binding to MmCpn is stochastic for each ATP occupancy level under low ATP concentration condition. As the ATP binding follows the stochastic manner between each two adjacent ATP occupancy levels, with the preset unique ATP distributions at each ATP occupancy level, it is reasonable to assume that the percentage for each distribution in each occupancy level does not change with respect to the change of the ATP abundance in the environment. Therefore, we expect that, under high ATP concentrations, ATP binding to MmCpn will still be stochastic at the occupancy levels we analyzed here but results in a higher average ATP occupancy. Of note, our study demonstrates that the intra-ring ATP binding is stochastic up to 6 ATPs and inter-ring binding is stochastic, thus the conclusion that ATP binding is stochastic and lack of KNF or MWC cooperativity can hold at the occupancy level not beyond 12 ATPs, it is possible that significant conformational change happens within and across the ring when more than 13 ATPs are bound to a single MmCpn. and therefore results in large cooperativity in the further binding (14–16). Such phenomenon would be consistent with the high ATP binding affinity in GroEL observed at high ATP occupancy reported in the previous study[17]. Therefore, the ATP binding at ATP occupancy higher than 13 ATPs remains to be tested for allostery. Our biochemical assay (Supplementary Fig. 1c) shows that at high ATP concentration where MmCpn molecules are saturated, hydrolysis could very well be cooperative.

The heterogeneous and non-concerted conformational change within the chaperonin MmCpn ring raises the possibility that the chaperonin could incompletely encapsulate substrate proteins during folding, which may facilitate the folding of protein substrates larger than its chamber volume. The different subunit conformations observed in our dataset may be interpreted as the trajectory of the ATP-induced subunit conformational change. Of note, our analysis suggests that molecular dynamics studies of ATP binding by MmCpn can be simplified to some extent to focus on a single subunit instead of the complex. Combining the heterogeneous data we obtained here together with the high resolution structures obtained for the closed MmCpn state[47] can provide very useful information for future molecular dynamics studies to explain how ATP energy transfer powers the MmCpn subunit conformational change.

Our study employs cryoEM to obtain a statistically significant measurement of the allosteric regulation of ATP binding in oligomeric chaperonin at near atomic resolution level. As a result, our study is a previously underexplored application of cryoEM single particle analysis to extraordinarily heterogeneous oligomeric complexes upon ligand binding, contrasting with many current researches pursuing one or a few structures after massive averaging subpopulations of identical particles and/or manipulation of the conditions to purposely focus on one or a few conformations. Our study demonstrates how to apply cryoEM single particle analyses to determine particle to particle conformational changes involving extraordinary heterogeneity in complexes made of multiple structurally quasi-identical subunits at high resolution. Similar application of cryoEM single particle analyses can shed light on allosteric transition for many additional oligomeric complexes to understand the biochemical process involving asynchronous structural variations of the molecular components at a single particle level.

## Methods

### Purification of MmCpn protein
Assembled complexes were purified following established protocol[33]. In brief, MmCpn was expressed off pET21a containing E. coli Rosetta DE3 pLys (EMD Millipore). After harvesting, cells were lysed by high pressure homogenizer on ice in MQ-A (20 mM HEPES-KOH pH 7.4, 50 mM NaCl, 5 mM MgCl$_2$, 0.1 mM EDTA, 10% (v/v) glycerol, 1 mM DTT, 0.1 mM PMSF) with fresh protease inhibitors. The cellular debris was cleared by $18,000 \times g$ centrifugation for 30 min and the supernatant was precipitated using 55% ammonium sulfate (0.326 g/ml) and followed by pelleting at $18,000 \times g$ for 30 min. The supernatant was then dialyzed against 2 L of MQ-A overnight before loading onto Q Sepharose FF. Proteins were eluted using a linear gradient from 0.05 to 1.0 M NaCl. Fractions containing MmCpn were diluted 1:1 with MQ-A before loading onto Hi-Trap Heparin equilibrated in MQ-A with 0.2 M NaCl and eluted with a linear gradient to 1.0 M NaCl. Chaperonin containing fractions were concentrated using an Amicon 10 K concentrator (Millipore) before loading onto gel filtration column Superose 6 10/300 GL. Fractions were analyzed by SDS-PAGE and fractions containing MmCpn were combined and concentrated using Amicon 10 K concentrators.

### Proteinase K protection assay
For proteinase K protection assay, 0.25 μM MmCpn in ATPase buffer was mixed with water (no ATP), with ATP (1 mM) or ATP•AlFx (1 mM ATP, 1 mM Al(NO$_3$)$_3$, 6 mM NaF). The reactions were incubated for 10 min at room temperature, followed by proteinase K (20 ng/μl) treatment at room temperature for 5 min. The digestion was stopped with 10 mM PMSF and analyzed by SDS-PAGE.

### Enzyme coupled ATPase assay
The ATPase activity of MmCpn (100 nM hexadecamer) at different ATP concentrations was determined using a NADH-coupled enzymatic assay. In brief, MmCpn in ATPase buffer (100 mM KCl pH = 7.4, 30 mM Tris-HCl, 5 mM MgCl$_2$, 10% Glycerol, and 1 mM DTT) was supplemented with 1 mM phosphoenolpyruvate, 5 units pyruvate kinase, 5 units lactate dehydrogenase, and 0.15 mM NADH. Then the reaction were added to ATP at different concentrations and the ATPase activity is detected by the absorbance at 340 nm at 25 °C.

### Rhodanese folding assay
Rhodanese was denatured in 6 M guanidinium-HCl with 5 mM DTT at 25 °C for 1 h. Denatured rhodanese was then diluted 100x into MmCpn with ATPase buffer (100 mM KCl pH = 7.4, 30 mM Tris-HCl, 5 mM MgCl2, 10% Glycerol, and 1 mM DTT) to final concentration of 1 μM. The final concentration of MmCpn is 0.25 μM. After 10 min incubation, the aggregated rhodanese was removed by centrifugation at max speed. Refolding was initiated upon addition of ATP (5 mM). Endpoint measurements were taken at 20 min and 60 min incubation at 37 °C. The yield of folded rhodanese was calculated by an enzymatic assay of rhodanese followed by the comparison with the activity of 1 μM of native rhodanese.

### CryoEM sample preparation and imaging
MmCpn and ATP were both incubated on ice. MmCpn was mixed with ATP (molar ratio of 1 MmCpn hexadecamer to 8 ATP molecules). The final MmCpn concentration and ATP concentration are 4 μM and 32 μM, respectively. In total, 3 μL MmCpn/ATP reaction solution was applied to glow discharged Quantifoil grid (R2/1) immediately after mixing. The grid was plunged into liquid ethane after 2 s blotting on Vitrobot (Thermo Fisher). Grid was then transferred into liquid nitrogen and loaded to Titan Krios (Thermo Fisher) for imaging.

Sample of MmCpn under ATP condition was imaged on Titan Krios (300KV) electron microscope (Thermo Fisher) with 20 eV energy filter with EPU v1.10. Images were recorded in K2 summit direct electron detector (Gatan) at defocus −0.8 to −2.0 micron with pixel size of 1.08 Å on specimen. A movie stack of 50 frames were collected at each specimen area with 10 s exposure at dose rate of 5 e⁻/Å²/sec on the detector.

### Image processing
In total, 6500 micrographs were motion-corrected and dose weighted using MotionCor2 v1.0.0[48] and then CTF-corrected using Gctf v1.06[49]. Around 900,000 particles were auto-picked using RELION v3.0.1[50]. Several rounds of reference-free 2D classification were performed to remove false positives based on their 2D class averages. The particle images were further subject to C8 symmetry-imposed 3D classification with reference generated from a low pass filtered MmCpn map. Individual classes from this 3D classification step were carefully examined. Particles of preferred orientation are discarded based on the abnormal 3D features and confirmation from reference-free 2D classifications. Several rounds of "3D–2D classification" procedure were repeated to reject particles of preferred orientation. In the end, ~160,000 particles were used for the final step of auto-refine 3D reconstruction with D8 symmetry imposed. The reconstructed map was corrected for the modulation transfer function of the detector and was sharpened by applying negative B-factors estimated by an automated procedure in RELION. The overall resolution was determined to be 3.9 Å based on the gold-standard criterion using an FSC of 0.143, and the local resolution was estimated by using MonoRes[51].

### Focused classification
For single ring analysis, the star file of the related particles was extended 2 times by relion_particle_symmetry_expand with D1 symmetry to cover the orientation for each of two rings in each particle. 3D focused classification of the rings was performed with C8 symmetry imposed but without orientational search. The workflow of subunit analysis is very similar to ring

analysis. First, the rings belonging to class 2 were selected and the star file of the related rings were extended 8 times by relion_particle_symmetry_expand with C8 symmetry to cover the orientation for each of eight subunits in a ring. Then 3D focused classification of the subunits was performed with a single subunit mask but without symmetry and orientational search. The mechanism for partial signal subtraction is illustrated in Supplementary Fig. 6a. The subtracted subunit sub-particles are generated using RELION[50].

**Model building and refinement**. The model of MmCpn subunit in the closed form (PDB ID: 3KFB) was fit into a single subunit density of auto-refined 3.9 Å map by rigid body fitting with Fit in Map tool from Chimera v1.14[52]. This subunit model was further refined with phenix.real_space_refine in Phenix v1.18.1, ISO-LDE v1.1.0[53] and then D8 symmetry model was generated by phenix. dock_in_map[45]. Models for different subunit conformations were generated by first rigid body fit into each density map then refined by phenix.real_space_refine program and ISOLDE.

All of the models are validated by phenix.validation_cryoem (Supplementary Tables 1, 2, 3, 4, 5). Difference maps between subunit density and map calculated from the model of protein only are generated by phenix.real_diff_map. The figures of the difference map are generated by Pymol v2.3.2[54] with the same contour level. All other figures are generated by Chimera.

**Reporting summary**. Further information on research design is available in the Nature Research Reporting Summary linked to this article.

## Data availability

The 3D cryoEM density maps of single subunit from conformation 1 to 8 and D8 symmetry averaged map are deposited in the Electron Microscopy Data Bank (EMDB) conformation 1: EMD-24324, conformation 2: EMD-24325, conformation 3: EMD-24326, conformation 4: EMD-24327, conformation 5: EMD-24330, conformation 6: EMD-24329, conformation 7: EMD-24331, conformation 8: EMD-24328, D8 averaged map: EMD-24363. The corresponding atomic coordinates are deposited in the Protein Data Bank, (PDB) conformation 1: 7R9H, conformation 2: 7R9E, conformation 3: 7R9I, conformation 4: 7R9J, conformation 5: 7R9O, conformation 6: 7R9M, conformation 7: 7R9U, conformation 8: 7R9K, D8 averaged model: 7RAK. Source image data are deposited to EMPIAR (EMPIAR-10770). Source data are provided with this paper.

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

## Acknowledgements

We thank Dr. Tom Lopez for providing E. coli strain pellets and protocols for protein purification. We thank Dr. Soung-Hun Roh for helpful discussion. We acknowledge support from NIH grants P01NS092525, P41GM103832, GM07407411 and S10OD021600.

## Author contributions

Y.Z. and W.C. conceived the project and approach. Y.Z. planned the experiments, collected and analyzed the data. Y.Z. took the lead in writing the manuscript. M.S., J.F., W.C. provided critical feedback and helped shape the research, analysis and manuscript.

## Competing interests

The authors declare no competing interests.
