## [Peer Review File · Nature Communications]

REVIEWER COMMENTS

Reviewer #1 (Remarks to the Author):

Estimating the relative populations of co-existing species with different numbers of a bound ligand can provide much insight into allostery or the lack of. Previously, this has been achieved using native mass spectrometry and single-molecule techniques. In this work by Chiu and co-workers, it is achieved for the first time, to the best of my knowledge, using cryo-EM single-particle analysis. Using this approach, the authors provide evidence that mmCpn does not bind ATP cooperatively. This unexpected finding is reconciled with previous reports (ref. 13) that mmCpn is allosteric by arguing that the difference is due to cooperativity in binding (measured here) vs. cooperativity in hydrolysis (measured before). There may, however, be alternative explanations as detailed in some of the comments below, all of which should be addressed.

1. Bottom of p. 3 – the difficulty in distinguishing between cooperativity in binding or hydrolysis is not due to the models but rather to the fact that hydrolysis, which is measured, is often assumed to be a proxy for binding. It is possible, however, to measure binding or hydrolysis directly as done, for example, in ref. 17 for binding, and then this uncertainty is circumvented.
2. The authors analyzed only 45% of the data, which is fine if these data are representative of the entire data. If, however, the analyzed data are biased in favor of certain states then the conclusions reached in this paper may be wrong.
3. The images were subjected to D8 symmetry averaging. It would be helpful to state more clearly why nucleotide occupancies are not averaged out during the data analyses.
4. Figure 4 – the agreement between the prediction and experiment for every binding number (e.g. 2, 3 etc.) is consistent with the homo-oligomeric nature of mmCpn but does not indicate whether binding is cooperative or not. Hence, the statement in the Abstract that “The lack of apparent correlation between adjacent subunits demonstrates the absence of cooperativity...” is wrong. The test for cooperativity is whether the distribution in panel A deviates from the binomial distribution (see Eq. S6 in ref. 17). My calculation for these data is that the Hill constant is indeed equal to 1 for that particular ATP concentration (i.e. there is no intra-ring cooperativity as stated by the authors). The Hill constant is, however, concentration-dependent so this conclusion remains tentative until it is tested for other concentrations. This may entail much extra work so a discussion of this point may suffice. A similar calculation should be done for the inter-ring allostery. It should also be noted that the MWC model (unlike the KNF model) does not invoke interactions between neighboring subunits.

Minor comments;

1. Who is the author of Ref. 3?
2. It should be made clear in the introduction that mmCpn is homo-oligomeric.

Reviewer #2 (Remarks to the Author):

Chaperonins are an important and ubiquitous group of molecular chaperones structurally characterised by a double-ring, oligomeric structure. There are two different groups: group I, found in eubacteria and in endosymbiotic organelles, and group II, found in Archaea and in the eukaryotic cytosol. The subunits that build the oligomeric structure have a common organization with three domains: the equatorial domain, responsible of the signalling originated from ATP binding and hydrolysis, which is transmitted to the other subunits of the ring and to the opposite ring; the apical domain, mainly responsible of the client protein recognition; and the intermediate, which transmits the signals originated in the nucleotide binding site to the apical domain. These set of signals are responsible of the large conformational changes that take place during the chaperonin cycle and that result in the two main conformations, the open that recognises and traps the client protein, and the closed one where folding occurs.

Whereas different kinetics studies have clearly shown for group I chaperonins (mostly for GroEL from *E. coli*) a positive cooperativity within the intra-ring subunits and a negative one between the two

rings, the results are not so clear with respect for group II chaperonins, in particular with regard to ATP binding.

Zhao et al. have used cryoelectron microscopy and image processing to address this issue in the archaeal chaperonin (thermosome) from *Methanococcus marimopaludis*, by determining the ATP occupancy and distribution within the subunits of each ring. By doing so, they find that ATP binds randomly to the thermosome subunits both within a ring and with regard the opposite one, which make the authors to conclude that there is no cooperativity in ATP binding in this kind of chaperonins.

- There are several concerns regarding the manuscript, some of them have to do with the data processing. Why do the authors use D8 symmetry during the first steps of the reconstruction process if they want to analyse the asymmetric nature of the chaperonin? Other groups, including that of one of the authors, have not applied symmetry when working with the asymmetric group II chaperonin TRiC/CCT, and have obtained high-resolution data (see for instance ref. 27 in the paper). I know that the asymmetry here could be even more complex but surely a larger set of data would help to overcome this problem. It is also surprising the low resolution of the reconstruction even when applying D8 symmetry, which does not allow for a proper reconstruction of the apical domains (see Fig. 1A and B). Some of the classes (Fig. 2C) only reveal a small part of the apical domain, so I find a bit daring to create an atomic model out of these.

This reinforces the need of acquiring a much larger set of data, including the use of other ATP concentrations to make a more comprehensive study.

- If the authors want to study the effect of ATP binding in the allostery of this thermosome, why they have used ATP instead of a non-hydrolyzable analogue? They acknowledge that there could have been hydrolysis and I agree with them.

The paper is an interesting one and the final conclusion (ATP binding to the thermosome subunits takes place in a stochastic manner) probably correct, but in my opinion this has been precariously obtained. This work does not deserve to be published in *Nature Communications* in its current form.

Reviewer #3 (Remarks to the Author):

The study by Zhao et al fundamental question relating to proteostasis. The manuscript addresses the mechanism of the opening and closing cycles of chaperonins (CPNs). These large double-ring oligomeric complexes enclose chambers in each ring that open and close in a highly cooperative and concerted reaction mediated by ATP binding and hydrolysis. The CPNs are divided into Group I (GroE) complexes that have a detachable lid (GroES) and Group II CPNs with a built in lid. The catalytic cycle of Group I CPNs has been intensely studied and characterized, whereas the Group II CPN cycling mechanisms are less studied. Conventional kinetic studies of both Group I and Group II CPNs reveal that both complexes have both positive and negative cooperativity of ATP hydrolysis. In this manuscript Zhao et al describe a novel Cryo-EM method for dissecting the initial binding of ATP to the Group II chaperonin hexadecamer from events following ATP hydrolysis through direct observation of particles frozen from a mixture of conformers partially saturated with ATP. By selecting conformers that are partially saturated the progression of binding could be mapped and turns out to be non-cooperative in that the classes of particles with neighboring ATP occupied subunits are not enriched statistically. This is a novel finding since it contrasts ATP binding with ATP hydrolysis reactions that are highly cooperative. It appears that the same stochastic binding process extends to inter-ring ATP binding indicating that the negative interring cooperativity observed for ATP hydrolysis does not apply to ATP initial binding.

This study relies entirely on the image processing expertise developed in the CryoEM field. In this case the image class selected for consideration of ATP occupancy is the class showing the best resolution. The major criterion for selection is the resolution of the apical domain, which is very poor (greater than 10 Angstroms) compared with the other domains. A previous study on mmCpn (Zhang et al, *Structure*. 2011 May 11; 19(5): 633–639.2011) used a lidless variant and the resolution was much better. There are a couple of control experiments that would increase confidence in the data collection strategy. One would be a separate Cryo-EM dataset with mmCpn saturated with ADP to visualize a fully open preparation to determine if there is a homogeneous preparation of native

structures. A second would be to measure the specific activity of ATPase activity of the preparation used for imaging.

These controls would increase the confidence that the preparation used was fully active and homogeneous, to guard against the possibility of selecting a subset of oligomers that were defective in ATP binding or in allosteric responses.

The paper is clearly written and organized, with very few grammatical errors. Line numbers would be helpful, to locate these typos.

Page 3, line 12. Delete An

Page 5 line 5. correct "each mmCpn subunits" to read "subunit"

Responses to Reviewers Comments on Manuscript NCOMMS-20-13414-T

Reviewer #1 (Remarks to the Author):

Estimating the relative populations of co-existing species with different numbers of a bound ligand can provide much insight into allostery or the lack of. Previously, this has been achieved using native mass spectrometry and single-molecule techniques. In this work by Chiu and co-workers, it is achieved for the first time, to the best of my knowledge, using cryo-EM single-particle analysis. Using this approach, the authors provide evidence that mmCpn does not bind ATP cooperatively. This unexpected finding is reconciled with previous reports (ref. 13) that mmCpn is allosteric by arguing that the difference is due to cooperativity in binding (measured here) vs. cooperativity in hydrolysis (measured before). There may, however, be alternative explanations as detailed in some of the comments below, all of which should be addressed.

1. Bottom of p. 3 – the difficulty in distinguishing between cooperativity in binding or hydrolysis is not due to the models but rather to the fact that hydrolysis, which is measured, is often assumed to be a proxy for binding.

Response

This is a good point and in page 3 (line 84-86) we revised the statement more accurately to clear the confusion.

It is possible, however, to measure binding or hydrolysis directly as done, for example, in ref. 17 for binding, and then this uncertainty is circumvented.

Response

We're aware that the nucleotide binding in chaperonin has been studied previously by native mass spectrometry as in ref 17 (doi:10.1073/pnas.1302395110) and single molecule fluorescence technique (doi:10.1073/pnas.1112244108). Both techniques can track the number of nucleotides in each chaperonin molecule. To our best knowledge, however, previous studies using these techniques do not address the questions regarding the distribution of nucleotides in each chaperonin. In our study, we are interested in determining if the different allosteric modes (MWC or KNF) exist and how they relate to cooperativity. To answer this question, we use cryo-EM single particle analysis in this paper to track not only the number but also the distribution of nucleotides in each chaperonin molecule. Furthermore, we investigate whether binding to one subunit enhances the likelihood of binding to a proximal subunit in the same complex. By examining a large number of mmCpn with different numbers of nucleotides occupancy, we conclude that homo-oligomeric chaperonin mmCpn binds ATP in a stochastic manner both within the ring and across the ring.

2. The authors analyzed only 45% of the data, which is fine if these data are representative of the entire data. If, however, the analyzed data are biased in favor of certain states then the conclusions reached in this paper may be wrong.

Response

We understand the concern from the reviewer about the data bias and we explain better the rationale for our data selection (line 155-157 and 176-178). The classification process eliminated those particles which lost structural integrity. These particles could be interpreted as damaged chaperonins either in solution or during the freezing step. Our only assumption to select data for further analysis is to assume that the functionally active particles have to maintain their structural integrity. Therefore, our analysis reflects those structurally intact (i.e. functional) chaperonins within which we found structural heterogeneity, which is a fair assumption. Based on our structural observations, we interpreted them through statistical arguments that led to the proposed hypothesis. Our logical reasoning convinced us that our analyses and conclusions are not biased. On the contrary, the image processing result based on the best practice in the field is surprisingly consistent with the statistical prediction.

3. The images were subjected to D8 symmetry averaging. It would be helpful to state more clearly why nucleotide occupancies are not averaged out during the data analyses.

Response

We appreciate the suggestion from the reviewer that we should state more clearly why we can observe nucleotide density in the D8 symmetry averaged map. The D8 symmetry was the first step of our analysis. The density signal from the nucleotide is strong probably because the nucleotide is less flexible in the binding pocket. Although many chaperonin subunits are empty without nucleotide occupancy, as shown in the second stage of analysis when we examine the structure of single subunits, the large number of nucleotide occupied subunits (roughly half of them) contribute enough signal that it is not completely averaged out in the initial D8 symmetry average map.

4. Figure 4 – the agreement between the prediction and experiment for every binding number (e.g. 2, 3 etc.) is consistent with the homo-oligomeric nature of mmCpn but does not indicate whether binding is cooperative or not. Hence, the statement in the Abstract that “The lack of apparent correlation between adjacent subunits demonstrates the absence of cooperativity...” is wrong. The test for cooperativity is whether the distribution in panel A deviates from the binomial distribution (see Eq. S6 in ref. 17). My calculation for these data is that the Hill constant is indeed equal to 1 for that particular ATP concentration (i.e. there is no intra-ring cooperativity as stated by the authors). The Hill constant is, however, concentration-dependent so this conclusion remains tentative until it is tested for other concentrations. This may entail much extra work so a discussion of this point may suffice. A similar calculation should be done for the inter-ring allostery. It should also be noted that the MWC model (unlike the KNF model) does not invoke interactions between neighboring subunits.

Response

We appreciate the reviewer's comments that "The lack of apparent correlation between adjacent subunits demonstrates the absence of cooperativity..." is wrong. And we realize and appreciate the reviewer's reminder that KNF model invokes interactions between neighboring subunits as ligand binds while the MWC model does not. After careful consideration, we revise our statement in the abstract to be more accurate as follows. "The lack of cooperativity between adjacent subunits in the chaperonin molecule suggests the absence of the sequential KNF allosteric mode in ATP binding step. The very heterogeneous non-concerted conformational change due to random ATP binding in the molecule also suggests the lack of concerted cooperativity by MWC model."

We agree with the reviewer that "the test for cooperativity is whether the distribution in panel A deviates from the binomial distribution (see Eq. S6 in ref. 17)". And we are glad to see that the reviewer's calculation that the Hill constant is 1 at the concentration under the study (i.e. there is no intra-ring cooperativity) agrees with our finding. We also agree with the reviewer that the Hill constant is concentration dependent. Therefore, our conclusion would be tentative until it is tested for other concentrations. We appreciate that the reviewer thinks that a discussion of this point would suffice considering much extra work. We discuss this further in the page 14 as follows.

"Our analysis under low ATP concentration shows that ATP binding to mmCpn occurs in a stochastic manner. We find that intra-ring ATP binding is stochastic up to 6 ATPs and inter-ring binding is stochastic, too. Neither KNF nor MWC cooperativity is observed in our study under the current condition. However, our conclusion that ATP binding is stochastic and lack of KNF or MWC cooperativity can hold at the occupancy level not beyond 12 ATPs. In fact, this observation of lack of cooperativity in ATP binding at low to intermediate occupancy (12 ATPs) is consistent with the observed low ATP binding affinity in GroEL at the similar occupancy level from reference 17 (doi:10.1073/pnas.1302395110). As the average ATP occupancy will shift towards the higher level as ATP concentration increases, it's possible that significant conformational change happens within and across the ring when more than 13 ATPs are bound to a single mmCpn, and thus results in large cooperativity when 14 to 16 ATPs are bound. Such phenomenon would be consistent with the high ATP binding affinity in GroEL observed at high ATP occupancy reported in reference 17 (doi:10.1073/pnas.1302395110).

Minor comments;

1. Who is the author of Ref. 3?

Response

Geoffrey M. Cooper.

2. It should be made clear in the introduction that mmCpn is homo-oligomeric.

Response

Done in the abstract and page 4 line 102.

Reviewer #2 (Remarks to the Author):

Chaperonins are an important and ubiquitous group of molecular chaperones structurally characterised by a double-ring, oligomeric structure. There are two different groups: group I, found in eubacteria and in endosymbiotic organelles, and group II, found in Archaea and in the eukaryotic cytosol. The subunits that build the oligomeric structure have a common organization with three domains: the equatorial domain, responsible of the signalling originated from ATP binding and hydrolysis, which is transmitted to the other subunits of the ring and to the opposite ring; the apical domain, mainly responsible of the client protein recognition; and the intermediate, which transmits the signals originated in the nucleotide binding site to the apical domain. These set of signals are responsible of the large conformational changes that take place during the chaperonin cycle and that result in the two main conformations, the open that recognises and traps the client protein, and the closed one where folding occurs.

Whereas different kinetics studies have clearly shown for group I chaperonins (mostly for GroEL from *E. coli*) a positive cooperativity within the intra-ring subunits and a negative one between the two rings, the results are not so clear with respect for group II chaperonins, in particular with regard to ATP binding.

Zhao et al. have used cryoelectron microscopy and image processing to address this issue in the archaeal chaperonin (thermosome) from *Methanococcus marimopaludis*, by determining the ATP occupancy and distribution within the subunits of each ring. By doing so, they find that ATP binds randomly to the thermosome subunits both within a ring and with regard the opposite one, which make the authors to conclude that there is no cooperativity in ATP binding in this kind of chaperonins.

- There are several concerns regarding the manuscript, some of them have to do with the data processing. Why do the authors use D8 symmetry during the first steps of the reconstruction process if they want to analyse the asymmetric nature of the chaperonin?

Response

The archaeal chaperonin mmCpn we study in this paper is homo-oligomeric. The initial step using D8 symmetry is a good approximation of the averaged structure that would allow us to determine the coordinates of each chemically identical subunit in each single mmCpn particle. This information is needed to perform the 3D focused classification in the subsequent steps.

Other groups, including that of one of the authors, have not applied symmetry when working with the asymmetric group II chaperonin TRiC/CCT, and have obtained high-resolution data (see for instance ref. 27 in the paper).

Response

We appreciate the question from the reviewer that D8 symmetry instead of C1 symmetry is applied in the first steps of the reconstruction process. We have two reasons for preferring D8 symmetry reconstruction strategy for the initial step here.

- 1) First, the hetero-oligomeric TRiC is made up of eight compositionally distinct subunits and is an absolutely different structure from mmCpn. Therefore, one cannot apply D8 symmetry analysis for TRiC structure determination. In contrast, the homo-oligomeric mmCpn consists of a single type of subunit, and D8 symmetry has been applied in many previous studies (doi: 10.1038/nature08701; doi: 10.1016/j.cell.2010.12.017).
- 2) Second, when C1 symmetry is applied to obtain an averaged map, one still assumes that each particle is structurally identical but this assumption is not true as shown in this analysis of mmCpn. We started with the assumption that each mmCpn particle may be structurally different because each compositionally identical subunit might assume different conformations with or without ATP under the ATP condition in our study. We agree that it is in principle possible to start with C1 symmetry in our initial step of image processing. However, we were concerned that if one used asymmetry to start with, we would need 16 times more particles to yield the same resolution in the first step, and there may be more errors to extract each subunit for the subsequent in-depth analysis of the conformation of each subunit if a lower resolution map was used.

In sum, there is no advantage to assume C1 symmetry to pursue our study not assuming each particle conformationally identical in the initial step of our image analysis. The C1 reconstruction alone will not yield the results of subunit heterogeneity, either. This is demonstrated by our previous analysis of GroEL consisting of 14 identical subunits, where we initially used D7 symmetry to obtain an averaged structure, followed by focused classification (doi:10.1073/pnas.1704725114). In the subsequent step of image processing, we relaxed the assumption of strict 7-fold symmetry and each subunit was found to adopt slightly different conformations. In both GroEL and the current mmCpn studies, we imposed symmetry at the beginning to obtain a highest possible resolution structure, which allows us to computationally extract each subunit for each particle accurately, this is then followed by subsequent steps of structural classification. Interestingly, we investigated the exact conformational heterogeneity of each particle, and found it to be different from particle to particle, as concluded here.

I know that the asymmetry here could be even more complex but surely a larger set of data would help to overcome this problem. It is also surprising the low resolution of the reconstruction even when applying D8 symmetry, which does not allow for a proper reconstruction of the apical domains (see Fig. 1A and B). Some of the classes (Fig. 2C) only reveal a small part of the apical domain, so I find a bit daring to create an atomic model out of these.

Response

The D8 symmetry averages out all the heterogeneity, which is more prominent in the apical domain. Therefore, it is not surprising that applying D8 leads to low resolution for the intermediate and the apical domains which have higher structural heterogeneity. The apical domain feature indeed appears only when the display threshold is set to a low-value (see Supplementary Figure 4). ATP binding induces a continuous conformational change, and 3D classification algorithms at present can only separate structural conformations into discrete conformations. Thus, a certain level of heterogeneity can persist in each subunit class and this results in lower resolution and lower threshold in the apical domain compared to the higher resolution feature in the equatorial domain, as the apical domain undergoes the largest movement among all three domains within the subunit upon ATP binding. In Figure 2C, we purposely used a high threshold in order to display the features in the ATP binding pockets but had to sacrifice the density displayed in the apical domain. However, to clarify the non-uniform nature of our map, subunit maps in lower threshold are also shown in our new Supplementary Figure 4 to show the presence of density in the apical domain.

We performed focused 3D classification at single subunit level to determine the conformational heterogeneity under the ATP condition used in this study. We observe eight different conformations at medium resolution (4-6Å) for each conformation (Figure 2B, Figure 3). Although we don't resolve side chains at this resolution, the current resolution for each conformation is good enough to distinguish the occupancy in the nucleotide binding pocket of each conformation and trace the peptide chain. Using the high resolution structure from previous studies as a reference, we are able to fit a model into each conformation since we observe features of backbone resolved in particular at the binding pocket.

This reinforces the need of acquiring a much larger set of data, including the use of other ATP concentrations to make a more comprehensive study.

Response

Although the map we obtained has an average resolution of 3.9 Å, we could see much higher resolution features, e.g. the nice sidechain density in the equatorial domain (Fig 1C). We expected to observe low resolution at the apical domain in the reconstruction map (see Fig. 1A and B) for two reasons. First, apical domain is naturally flexible in solution when mmCpn adopts the open state and this was observed in low resolution in multiple reported cryo-EM structures. Secondly, we purposely create even higher conformational heterogeneity by using low concentration of ATP and aim to observe and analyze this heterogeneity in our study. Therefore, the large conformational change (up to 35Å as shown in Figure 2C) at the apical domain explains the expected low resolution in the D8 average map. We would expect to improve the resolution in the equatorial domain by collecting a larger dataset. However, we wouldn't expect to see the resolution at the intermediate and apical domain to improve in any D8 averaged map by doing so.

In order to study whether ATP binds to chaperonin MmCpn stochastically or cooperatively, we have to focus on not-fully ATP bound particles. At high ATP concentration, all the subunits would

be saturated with ATP anyway, giving no information about the cooperativity. Half-saturated ATP concentration has the most discriminatory power, which explains the mmCpn to ATP ratio we used in our study to observe ATP binding mechanism in those non-fully ATP bound particles. As stated above, the data analyzed was found to reveal examples of all the ATP occupancy levels, from zero to 16, in the expected ratios, hence the data analysis is not biased. And from our analysis of the relationship of each ATP-bound subunit to the other instances of ATP-bound subunits in every particle, we conclude that ATP binding is stochastic at each ATP occupancy level we examined. As the average binding in mmCpn at high ATP concentration will shift from low occupancy to high occupancy level gradually, we think the ATP binding might still be stochastic under other ATP concentrations. And we covered this in the discussion section of the manuscript. Also see our response to reviewer 1.

- If the authors want to study the effect of ATP binding in the allostery of this thermosome, why they have used ATP instead of a non-hydrolyzable analogue? They acknowledge that there could have been hydrolysis and I agree with them.

Response

Non-hydrolyzable analogues are not a natural substrate and differ from ATP exactly in the region of the nucleotide that induces the conformational change. Furthermore, AMPPMP and even more ATP-gammaS can be slowly hydrolyzed and may generate different products hard to pin down to be physiologically relevant. These analogues at high concentrations are good for stabilizing a rigid conformation state particularly for high resolution study to get a baseline structure. In this study, we are interested to examine the consequence of structural heterogeneity using a natural ligand at the expense of the very high resolution one would normally want.

The paper is an interesting one and the final conclusion (ATP binding to the thermosome subunits takes place in a stochastic manner) probably correct, but in my opinion this has been precariously obtained. This work does not deserve to be published in Nature Communications in its current form.

Response

We appreciate this reviewer to find this interesting and find our conclusion probably correct. We believe that the conclusion is based on solid and highly sophisticated analyses with a well validated pipeline and the natural chaperonin ligand. Based on the arguments above, we have carefully designed the experiments and unbiasedly analyzed the data using the best practice in the field of image processing. We don't think the final conclusion is precariously obtained, and it is indeed correct. **We note that this is to our knowledge, the first examination of single-particle cooperativity of such a complex enzyme solely using cryo-EM, which is an important advance for the field that will have a significant impact on the study of such complexes and many other enzymes.**

Reviewer #3 (Remarks to the Author):

The study by Zhao et al fundamental question relating to proteostasis. The manuscript addresses the mechanism of the opening and closing cycles of chaperonins (CPNs). These large double-ring oligomeric complexes enclose chambers in each ring that open and close in a highly cooperative and concerted reaction mediated by ATP binding and hydrolysis. The CPNs are divided into Group I (GroE) complexes that have a detachable lid (GroES) and Group II CPNs with a built in lid. The catalytic cycle of Group I CPNs has been intensely studied and characterized, whereas the Group II CPN cycling mechanisms are less studied. Conventional kinetic studies of both Group I and Group II CPNs reveal that both complexes have both positive and negative cooperativity of ATP hydrolysis. In this manuscript Zhao et al describe a novel Cryo-EM method for dissecting the initial binding of ATP to the Group II chaperonin hexadecamer from events following ATP hydrolysis through direct observation of particles frozen from a mixture of conformers partially saturated with ATP. By selecting conformers that are partially saturated the progression of binding could be mapped and turns out to be non-cooperative in that the classes of particles with neighboring ATP occupied subunits are not enriched statistically. This is a novel finding since it contrasts ATP binding with ATP hydrolysis reactions that are highly cooperative. It appears that the same stochastic binding process extends to inter-ring ATP binding indicating that the negative interring cooperativity observed for ATP hydrolysis does not apply to ATP initial binding.

This study relies entirely on the image processing expertise developed in the CryoEM field. In this case the image class selected for consideration of ATP occupancy is the class showing the best resolution. The major criterion for selection is the resolution of the apical domain, which is very poor (greater than 10 Angstroms) compared with the other domains. A previous study on mmCpn (Zhang et al, Structure. 2011 May 11; 19(5): 633–639.2011) used a lidless variant and the resolution was much better.

Response

We appreciate the reviewer's suggestion that a lidless variant helps improve the resolution. However, given that previous studies from our group showed that the lid contributed to the allosteric regulation of group II chaperonins (10.1038/nsmb1236), we chose to study wild-type mmCpn as we cannot rule out the possibility that the lid interaction between neighboring subunits in the relatively closed state might be one of the factors contributing to ATP binding cooperativity.

There are a couple of control experiments that would increase confidence in the data collection strategy. One would be a separate Cryo-EM dataset with mmCpn saturated with ADP to visualize a fully open preparation to determine if there is a homogeneous preparation of native structures. A second would be to measure the specific activity of ATPase activity of the preparation used for imaging.

Response

We appreciate the reviewer's suggestions, and we agree that the ATPase activity of the preparation can help to increase the confidence that the preparation used is fully functional and active. We included this assay as one of our approaches to address the concern in the following response. However, we consider the experiment of saturating mmCpn with ADP is not ideal to fulfill the purpose as shown in the previous paper that ADP binding will cause additional heterogeneity among molecules (doi:10.1016/j.str.2008.01.016).

These controls would increase the confidence that the preparation used was fully active and homogeneous, to guard against the possibility of selecting a subset of oligomers that were defective in ATP binding or in allosteric responses.

Response

We appreciate the suggestion from the reviewer that we should have the control experiment to validate that the preparation used was homogeneous and functional in order to guard against the possibility of selecting a subset of oligomers that were defective in ATP binding or in allosteric responses. We are fully aware of the reviewer's concerns and addressed them in two ways:

1) To prove that the preparation is fully active and functional, we performed a well validated proteinase K (PK) digestion assay to test that the preparation can reach the closed state in the presence of ATP. The sample from the same prep used for the cryo-EM study in apo state (no nucleotide bound) and in the presence of ATP and of ATP/AIFx was digested with PK showing that the entire population of chaperonin can reach the ATP induced closed state. The assay demonstrates the sample prepared is of high purity and fully closed under ATP condition (ATP state and ATP/AIF3). In addition, we also performed ATPase assay on mmCpn at high ATP concentration. We observe the expected enzyme turnover rate by monitoring the hydrolysis product Pi with quinaldine red-based phosphate assay. These data are included in our new Supplementary Figure 1.

2) We also performed single particle analysis with the same batch of chaperonin under both high ATP condition and ATP/AIFx (Unpublished). Under high ATP concentration, we observe that the majority of the particles reach fully closed state as expected from a fully functional preparation and as observed by previous researchers. As suggested from the study (doi: 10.1016/j.str.2011.03.005) that ATP hydrolysis is required to reach fully closed state, the observation of majority particles in their fully closed state under ATP condition indicates that the preparation is fully active and functional.

3) After the classification process, we eliminated the particles which lost structure integrity. These particles could be interpreted as damaged chaperonins either in solution or during the freezing step. It is fair to assume that the functionally active particles have to maintain their high resolution structures. Therefore, our analysis reflects those structurally intact (i.e. functional) chaperonins within which we found structural heterogeneity. Therefore, we do not think that our conclusion is biased. On the contrary, the image processing result based on the best practice in the field is surprisingly well explained from a statistical perspective (answer for 1st reviewer, 2nd question).

The paper is clearly written and organized, with very few grammatical errors. Line numbers would be helpful, to locate these typos.

Response

We are glad that the reviewers did not detect serious grammatical errors. We worked very hard to meet high standards of written language prior to submitting the paper for publication in a prestigious journal like *Nature Communications*. It is an excellent suggestion to include line numbers in our revision.

Page 3, line 12. Delete An

Done.

Page 5 line 5. correct "each mmCpn subunits" to read "subunit"

Done.

REVIEWER COMMENTS

Reviewer #1 (Remarks to the Author):

The revised version of the paper is improved. A few more points that should be addressed are listed below.

1. Lines 247-255: I would be more convinced by cases in which an open subunit is flanked by two closed ones (rather than a closed one flanked by two open ones). Do the authors see this and if not then why?
2. Lines 266-268 – similar frequencies of 1 or 2 nucleotides bound to a subunit pair does not indicate random binding. Random binding is indicated by a binomial distribution that includes the frequency of empty subunits as well and is a function of the overall fractional saturation (i.e. binding probability). If the fractional saturation is 0.5 then 25% of the pairs should be occupied at both sites, 25 % should be empty and 50% should be occupied at one site for random binding.
3. The issue of the D8 symmetry averaging raised by several reviewers remains unclear. If D8 symmetry averaging cannot be used for TRIC, as the authors state, because it is hetero-oligomeric then why can it be used for mmCpn which although homomeric is still asymmetric because of ATP binding? This part of the data analysis still needs somewhat better explaining.

Minor points:

1. Line 95 – Native mass spectrometry is not a single-molecule technique but it enables measuring distributions and not just averages.
2. Lines 120-121- sentence not clear. How can a single particle be seen to have 2 preferred orientations?
3. Line 307 – The colors don't really match those of Figure 2. I suggest using identical shades of each color.

Reviewer #2 (Remarks to the Author):

I maintain my concerns with regard to the manuscript.

My basic concern is the methodology used in the work, which is not properly explained in the manuscript (I don't think the process can be reproduced if one follows what is written in the paper. To my question of why the authors have not used C1 symmetry, the authors respond that "The archaeal chaperonin mmCpn we study in this paper is homo-oligomeric. The initial step using D8 symmetry is a good approximation of the averaged structure that would allow us to determine the coordinates of each chemically identical subunit in each single mmCpn particle. This information is needed to perform the 3D focused classification in the subsequent steps". I know that the mmCpn the authors study in this paper is homo-oligomeric, but they're looking for structural asymmetries, so they have to assume that they have a "hetero-oligomer". I strongly believe that the best approximation is still to assume C1 symmetry, select a few millions particles at 2-3 different nucleotide concentrations and carry out a thorough classification, something that is in the hands of such a good cryoEM group.

They later claim that "there is no advantage to assume C1 symmetry to pursue our study not assuming each particle conformationally identical in the initial step of our image analysis" and I'm afraid I cannot agree with this. If one assigns angles to particles that have a subunit in a different conformation, using as a model a structure with C8 or D8 symmetries (i.e. all the subunits are equal), there will be irremediably a random distribution of the angles of the subunit that experimentally is different. In other words, if the model has symmetry-imposed, 8 identical subunits, the computational alignment cannot assign the subunit that is experimentally different to any of those 8 units and therefore the angle assignment will be totally random as it is the case to the analysis that the authors carry out later. In fact, with this (erroneous) experimental protocol the only mathematically possible solution is the random distribution of the subunits with different conformation.

The lack of particles also affects the analysis of intra-ring ATP occupancy and distribution (Fig. 4). The authors describe 26 populations (27 if one considers the chaperonins with one ATP-containing subunit; the most populated of all possibilities). I would argue that 160,000 particles for a proper classification with all these classes. Again, more particles are needed.

Other points

- I don't understand why the 3D reconstruction shown in Fig. 1 is so poor, having reached 3.5 Å resolution and having imposed D8 symmetry. The apical domains are hardly visible, which is not the case for other type II reconstructions of similar resolution published by other groups.
- In Fig. 3 the authors show a difference map analysis of the nucleotide occupancy in each of the eight conformations. For that they show the atomic structure in the background and the density of the difference map. It would have been nice to have seen the whole difference map of this area and the sigma used.

Reviewer #3 (Remarks to the Author):

The paper has been revised in response to three reviews with clarification of some issues. One important issue is the quality of the CPN preparation and this has now been assessed by control experiments using the Proteinase K digestion protocol and by an ATPase hydrolysis assay shown in Supplementary Figure 1 A and B. Figure 1A does show, by the prevention of protease digestion, that the preparation is largely closed in the presence of ATP. However, there is still significant degradation, shown in Figure 1 A Lane 3. The assay shown in Fig 1B establishes linearity of the assay but **does not report on the specific activity of the CPN preparation**. Thus although there is functionality we cannot establish the quality of the preparation relative to other studies. The possibility that a significant proportion of inactive complexes are in the preparation still exists. The authors have not repeated experiments at variable ATP concentrations as recommended in the original reviews, leading to the weakening on their conclusion by the addition of the phrase "at the ATP condition that we study." (Line 108-109). They also chose not to use non-hydrolysable ATP analogs which would have been an additional assurance of functionality of the preparation by showing a fully closed condition in all of the images.

May 30, 2021

Point-by-point Responses to all the reviewers' comments for manuscript (NCOMMS-20-13414A)

Reviewer #1:

The revised version of the paper is improved. A few more points that should be addressed are listed below.

1. Lines 247-255: I would be more convinced by cases in which an open subunit is flanked by two closed ones (rather than a closed one flanked by two open ones). Do the authors see this and if not then why?

Response: We indeed observe cases where an open subunit is flanked by two closed ones. We now present a reconstructed map of three subunits in this particular conformational state and build a structural model of these subunits into the map. The map and model are shown in Supplementary Figure 3.

2. Lines 266-268 – similar frequencies of 1 or 2 nucleotides bound to a subunit pair does not indicate random binding. Random binding is indicated by a binomial distribution that includes the frequency of empty subunits as well and is a function of the overall fractional saturation (i.e. binding probability). If the fractional saturation is 0.5 then 25% of the pairs should be occupied at both sites, 25 % should be empty and 50% should be occupied at one site for random binding.

Response: This is an excellent point! We appreciate the Reviewer's correction here. The traditional definition of fractional saturation in biochemistry is the bulk concentration of ligand per binding site. If there is no allostery between rings, the reviewer is correct. Figure 5B of our original manuscript showed that for subunit pairs located across the equator from each other at this concentration of ATP, the average relative frequency of double site nucleotide occupancy is similar to that of the single site occupancy. We agree that this finding, *per se*, does not support the claim that the ATP binding is random between two rings at the subunit level. Therefore, we further analyzed these inter-ring subunit pairs, similar to the way we analyzed intra-ring subunits in Figure 4B, at different ATP occupancy levels of each particle (i.e., ATP fractional saturation for that particular particle). As shown in the new Figure 5B, the relative frequency for double ATP occupancy gradually increases and eventually surpasses the single occupancy as the fractional saturation of a particle increases. We compare the ATP distribution in inter-ring subunit pairs at different per-particle ATP fractional saturation levels to that of the random distribution, and we find that the difference between our observation and the random distribution prediction is quite small, especially at moderate to high per-particle fractional saturation (supplementary Figure 8). This analysis supports the conclusion that the ATP binding in the inter-ring subunit pairs is close to random. Thus, our analysis, at a single ATP concentration (fractional saturation of 0.5, by biochemical definition), reveals the whole spectrum of ATP fractional occupancy, from 0 to 16 ATP bound, both within and across rings.

3. The issue of the D8 symmetry averaging raised by several reviewers remains unclear. If D8 symmetry averaging cannot be used for TRiC, as the authors state, because it is hetero-oligomeric then why can it be used for MmCpn which although homomeric is still asymmetric because of ATP binding? This part of the data analysis still needs somewhat better explaining.

Response:

We apologize that our rationale of the data analysis was insufficiently explained in the original manuscript. In the revised text and Method section, we add a description of the problem at hand and of our solution to it. Furthermore, we provide a detailed rationale of our analysis in our response to the questions raised by reviewer 2 and also in the revised text on page 7.

Each ring of the TRiC is made of 8 distinct subunits, and we agree that it would require C1 symmetry to solve its structure. However, MmCpn is made of 8 chemically *identical* subunits. Accordingly, it has been shown to be feasible to solve its near atomic structure using D8 symmetry if we do not care about characterizing the single subunit heterogeneity (DOI: 10.1038/nature08701). The important point in our analysis is that for MmCpn + ATP, the equatorial domains are nearly structurally invariant and thus can be examined using D8 symmetry while the apical domains are so conformationally heterogeneous that they cannot be resolved even using C1 symmetry. This point is brought home with our new analyses described in the revised text: when we applied either D8 or C1 symmetry, we observed high resolution features in the equatorial domain, comprising 60% of the subunit volume, which would not be resolved if D8 symmetry were inappropriate. **In either symmetry assumption, the map densities in the apical domains are poorly resolved.** Hence, we developed a three stages analysis pipeline (more details in our response to reviewer 2), with a completely new way to analyze the conformational heterogeneity in the apical domains **for each individual particle**. The approach is aimed at gaining structural insight into chaperonin allostery using cryoEM.

Minor points:

1. Line 95 – Native mass spectrometry is not a single-molecule technique but it enables measuring distributions and not just averages.

Response: Corrected.

2. Lines 120-121- sentence not clear. How can a single particle be seen to have 2 preferred orientations?

Response: We apologize for the confusion here. In our cryo-EM experiment, we freeze the sample solution with chaperonin on the grid. Many MmCpn molecules in the solution are frozen at different views on the grid. However, due to the hydrophobicity at the apical domain, fully open particles will tend to adopt top views and tilted top views on the grid so that their apical domain can be closer to the water-air interface. These two views are redundant in the map reconstruction, therefore part of these views are excluded during the map reconstruction by the current algorithm as one of the commonly used solutions to the preferred orientation problem.

3. Line 307 – The colors don't really match those of Figure 2. I suggest using identical shades of each color.

Response: Thank you for the excellent suggestion, it is corrected.

Reviewer #2:

I maintain my concerns with regard to the manuscript.

Q2.1. My basic concern is the methodology used in the work, which is not properly explained in the manuscript (I don't think the process can be reproduced if one follows what is written in the paper).

To my question of why the authors have not used C1 symmetry, the authors respond that “The archaeal chaperonin MmCpn we study in this paper is homo-oligomeric. The initial step using D8 symmetry is a good approximation of the averaged structure that would allow us to determine the coordinates of each chemically identical subunit in each single MmCpn particle. This information is needed to perform the 3D focused classification in the subsequent steps”. I know that the MmCpn the authors study in this paper is homo-oligomeric, but they're looking for structural asymmetries, so they have to assume that they have an “hetero-oligomer”.

Response:

We regret that our approach might have not been explained clearly in the original manuscript. We now add a description of the goal of our investigation and our technical approach in the revised text and Methods section.

The aim of our investigation is to study allosteric conformational changes in a homo-oligomeric complex of a protein complex like MmCpn in response to the presence of ATP. Our approach is to use cryoEM with an advanced image processing tool to determine the **structure of each subunit of an individual MmCpn particle at an ATP concentration that is designed to produce a heterogenous population of ATP occupancy**. Our cryoEM study is intended to determine the locations of the ATP bound subunits in each individual particle.

We agree that we are looking for structural asymmetries. However, we did not assume that the kind of asymmetry in the particles is invariant from particle to particle. To cope with this challenge, we develop the following structural analysis pipeline:

- 1) In the first stage, we exploited the fact that the structure of the equatorial domains are relatively invariant. This is evidenced in our initial reconstruction using D8 symmetry. In view of the reviewer's suggestion, we now carry out the C1 symmetry reconstruction (Figure 1 below and new Supplementary figure 3). Interestingly, both C1 and D8 symmetry-imposed maps show high resolution features (alpha helices) indicating high structural homogeneity in the equatorial domains of all subunits (Figure 1B). On the other hand, as apparent in the side views of Figure 1A, both maps do not resolve the densities in the apical domain. These observations support our conclusions that D8 symmetry is strictly followed at the equatorial domain, and there is high conformational heterogeneity in the upper domains of individual subunits within each particle, as well as among different particles. **This new analysis supports our approach because it yields the same initial structure as that obtained by D8 symmetry which was used for subsequent analysis for each subunit in each particle.**

Figure 1. (A) B-factor sharpened maps of C1 (yellow) and D8 symmetry-imposed (cyan) reconstructions. (B) Segmented cryo-EM map for a single subunit from C1 reconstruction (yellow) and D8 symmetry-imposed (cyan) reconstructions in A. The equatorial domains of both maps show the alpha helices features whereas the apical domains remain poorly resolved due to conformational heterogeneity within and across particles. (C) FSC curve of masked C1 density map showing resolution of 4.3 Å according to the gold standard FSC (0.143 criterion).

- 2) In the second stage, we examined **the departure from symmetry** in the apical domain for every subunit of every individual particle, with a new classification protocol. We were able to classify **over one million subunits** into ten different classes. The number of subunits in each class is large enough to observe the individual subunit conformation and nucleotide occupancy at reasonable resolution (4-6 Å). This analysis revealed 8 major classes of apical domain conformations for the chemically identical subunits in all particles: 4 in the ATP-bound state and 4 in the unbound state. This type of conformational heterogeneity is within our expectation with multiple protein conformations **at subunit level**.
- 3) A unique aspect of our investigation is the third stage. Since RELION records the position of each conformational variant in each particle, we can trace back which subunits in each particle assume which conformations. **It is important to note that we are not searching for the structures of subpopulations of the particles that have identical conformations for all of their 16 subunits, but more importantly, for the locations of individual subunits with and without bound ATP in order to determine whether there was an allosteric effect on ATP binding (not hydrolysis). Our study demonstrates that such a question can be answered at a single particle level.** Figures 2B and 3 in our original manuscript showed that both nucleotide unoccupied subunits and nucleotide occupied subunits adopt four possible types of conformations each. We have shown in Figure 4B of our manuscript 26 of the 30 types of possible populations of ATP distribution, not including 0, 1, 7 or 8 ATP bound, each of which has only one permutation. **If each subunit can have different conformation, the total possible subunit conformations at the particle level amounts to 480 billion. In this**

probabilistic estimate, the chance of having any two particles with the exact same conformation at the subunit level is extremely low in our data. Our statistical analysis clearly indicates that the nucleotide binding step to Mm-Cpn occurs in a stochastic manner.

Our new data analysis clearly demonstrates that using C1 symmetry or D8 symmetry does not affect the initial models obtained, **which are the starting points for subsequent analyses for individual subunits in every asymmetric particle.** We hope this explanation clarifies the underlying logic of our analyses. We thank the reviewer for raising this question. Our revised manuscript is improved to explain the rationale of our approach to study this complex single particle problem. Regarding the replication of our methodology by other investigators, we hope that the revised text makes it clear the rationale which is new in this manuscript. In terms of reproducing our process, we assume familiarity with the RELION software package, which is daunting to use in its full functionality.

Q.2.2. I strongly believe that the best approximation is still to assume C1 symmetry, select a few millions particles at 2-3 different nucleotide concentrations and carry out a thorough classification, something that is in the hands of such a good cryo-EM group. They later claim that “there is no advantage to assume C1 symmetry to pursue our study not assuming each particle conformationally identical in the initial step of our image analysis” and I’m afraid I cannot agree with this.

Response: We disagree that our classification step is not thorough and rigorous. In any reconstruction study, there is a fundamental assumption that particles are similar enough to yield at least an averaged initial structure regardless of their symmetry. As explained above, there is no difference between using D8 and C1 in our initial step of data processing. We have a statistically adequate number of subunits from our particles for the downstream analysis of structural **heterogeneity at the subunit level.** We are not making any final conclusions on the structure of the particles with either symmetry assumption in the first step. Instead, we are using the information of the subunit position to facilitate identifying each subunit’s structure in each particle with the RELION classification protocol in the subsequent steps. **The ultimate purpose of our study is to show which subunit is occupied with ATP within each 16-mer oligomer and to this end we need to ultimately assess the conformation of each subunit in each particle.** This will allow us to understand the biochemical principle of allosteric regulation in protein and ligand interaction, i.e. Does ATP bind to MmCpn cooperatively or randomly? We are not trying to find a single (or even several) unique structures for subpopulations in the millions of identical 16-subunit particles, because they simply do not exist (and doing this would not answer our posed biochemical question). In fact, we are after a more interesting question: at concentrations where the particles have the most variability, what are the spatial relationships between ATP-bound vs. apo-subunits? This question goes to the heart of allosteric regulation. Therefore, our study is different from those generally reported in the literature with one or a few structures after massive averaging by manipulating the chemical condition to force all the apical domains to be conformationally more homogeneous, as suggested by the reviewer. We agree with the reviewer that it is of interest to study the MmCpn under different nucleotide and substrate conditions. Such investigations are beyond the scope of this manuscript.

Q.2.3. If one assigns angles to particles that have a subunit in a different conformation, using as a model a structure with C8 or D8 symmetries (i.e. all the subunits are equal), there will be irremediably a random distribution of the angles of the subunit that experimentally is different. In other words, if the model has symmetry-imposed, 8 identical subunits, the computational alignment cannot assign the subunit that is experimentally different to any of those 8 units and therefore the angle assignment will be totally random as it is the case to the analysis that the authors carry out later. In fact, with this (erroneous) experimental protocol the only mathematically possible solution is the random distribution of the subunits with different conformation.

Response: The reviewer is correct if we consider only one subunit out of 8 to be different. In that case, since the different subunit could be any one of the 8 subunits, all 8 possible locations for the different subunit would be equivalent. Then the assignment of that subunit would be random. However, in the case of more than one different subunit (even just two), there are **distinguishable** different arrangements in any given particle, (adjacent, skip one, skip two, and opposite, analogous to ortho-, meta-, and para- conformations of xylene, around the ring) as we show in Figure 4B. We actually found all these arrangements in our analyses, and all the rotational permutations thereof. Under such circumstances where both the number of different subunits and their arrangements vary tremendously among particles, the apical domain should not be expected to be resolved. For this reason, C1 symmetry application generates the same result as the D8 symmetry as now shown by our new analysis.

The novel and important aspect of our analysis pipeline is that for either D8 or C1 symmetry reconstructed map we derive the Euler angle orientation for each particle, from which we can deduce the “coordinate of each subunit” in the projection image readily and accurately. Then we classify each subunit’s structure, not the entire particle’s structure, using RELION, and we can classify the apical domain conformations into one of eight conformational classes. We averaged those **subunits** with identical structures and then mapped them back to which subunit and which particle has that conformation for their apical domain. This approach allows us to determine the nucleotide bound subunit distribution **within each particle** at different ATP occupancy levels. Based on these analyses we draw our conclusion of random nucleotide binding to subunits in MmCpn. Therefore, there is no bias in the process. We do not believe that we use wrong logic or an erroneous computational protocol. Again, we are not trying to find an average of an ensemble of particles with identical conformations using a symmetry imposition.

Q.2.4. The lack of particles also affects the analysis of intra-ring ATP occupancy and distribution (Fig. 4). The authors describe 26 populations (27 if one considers the chaperonins with one ATP-containing subunit; the most populated of all possibilities). I would argue that 160,000 particles for a proper classification with all these classes. Again, more particles are needed.

Response: We disagree with the reviewer on several points in the context of our data and the biological goal of our study. First, including the populations having 0, 1, 7 or 8 ATP-bound subunits (each of which, trivially, has only one distinct permutation, so is not shown in Figure 4), there are 30 populations represented here. Second, and most importantly, as is clearly shown in Figure 4, the chaperonins with one ATP-containing subunit are among the **least** populated of all possibilities, with the population in each ring having a peak at 5-6 ATPs per oligomer. **Our**

scientific question in this investigation is the distribution of ATP bound subunits in each ring and our method of analysis answers it. We took a tiered, bottom-up approach in our analysis by first classifying over one million individual subunits into ten different classes. The number of subunits in each class is large enough to observe the subunit conformation and nucleotide occupancy at reasonable resolution (4-6 Å) sufficient to answer our biological question. Then we trace the subunits in the different classes back to their locations in the particles they came from, and study the resulting compositional and conformational heterogeneity at the particle level. The statistical analysis clearly indicates that the nucleotide binding step to MmCpn occurs in a stochastic manner. We are not trying to derive a unique structure of one conformational arrangement in a single MmCpn. Therefore, more particles would not change the conclusions on the posed question.

Q.2.5. Other points

• I don't understand why the 3D reconstruction shown in Fig. 1 is so poor, having reached 3.5 Å resolution and having imposed D8 symmetry. The apical domains are hardly visible, which is not the case for other type II reconstructions of similar resolution published by other groups.

Response: We hope the explanation above clarifies why the apical domains are not resolved. The apical domains are too mobile and heterogeneous to allow resolution, while the equatorial domains are well resolved. **In the current study, we are interested in using chemical conditions that would lead to such different subunit conformations, since our underlying question is to capture allosteric changes.** None of the previous studies posed the same question as ours. We could have used a chemical condition to force all the apical domains to be conformationally identical. Indeed, we have published several cryoEM studies of MmCpn including the first atomic model of cryoEM single particle done before the resolution revolution and subsequently confirmed to be correct (doi: 10.1074/jbc.M110.125344) by X ray crystallography. Significantly, in those studies, ATP-AIFx was used to lock all the apical domains in the closed state and the particles follow excellent D8 symmetry, but this has no bearing on our question of which subunits bind ATP, and in what location around and across the 8-member rings, under sub-saturating ATP concentration.

Q.2.6. • In Fig. 3 the authors show a difference map analysis of the nucleotide occupancy in each of the eight conformations. For that they show the atomic structure in the background and the density of the difference map. It would have been nice to have seen the whole difference map of this area and the sigma used.

We have generated a new set of images for the difference maps in Figure 3. They show a more global view of the nucleotide binding domain. These difference maps are displayed at +4 sigma.

Reviewer #3 (Remarks to the Author):

The paper has been revised in response to three reviews with clarification of some issues. One important issue is the quality of the CPN preparation and this has now been assessed by control experiments using the Proteinase K digestion protocol and by an ATPase hydrolysis assay shown in Supplementary Figure 1 A and B. Figure 1A does show, by the prevention of protease

digestion, that the preparation is largely closed in the presence of ATP. However, there is still significant degradation, shown in Figure 1 A Lane 3. The assay shown in Fig 1B establishes linearity of the assay but **does not report on the specific activity of the CPN preparation**. Thus although there is functionality we cannot establish the quality of the preparation relative to other studies. The possibility that a significant proportion of inactive complexes are in the preparation still exists.

Response:

Reviewer 3 was concerned about our functional assays. In our revised manuscript, we presented a Proteinase K digestion assay, which is a more stringent diagnostic of functionality and should have addressed the Reviewer 3's concern regarding the sample quality. However, we may not have explained well the details of this assay in the revised manuscript, which may have resulted in the reviewer's misinterpretation of this assay. We explain in more detail below. **In addition, we have performed the ATPase measurements of the chaperonin requested by Reviewer 3, as well as an additional rhodanese folding assay, which further validate that our sample preparation is fully functional. With these three orthogonal assays, we believe that we have convinced the reviewer 3 that the MmCpn sample used in our experiment is functional.**

Firstly, in the **ATPase assay**, we determined the steady state hydrolysis rate of MmCpn under a range of ATP concentrations. As shown in Figure 2C and 2D, hydrolysis rate increases sharply at low ATP concentration and then drops slightly at high ATP concentrations *as seen in previous analyses of MmCpn allosteric regulation* (DOI:10.1038/nsmb1236). This experiment indicates that the MmCpn preparation used is fully active and functional and subject to proper allosteric regulation.

Secondly, we present a **rhodanese folding assay** that tests the ability of our MmCpn preparation to refold the denatured substrate protein rhodanese in an ATP-dependent manner. As shown in Figure 2B, MmCpn can efficiently fold rhodanese in an ATP dependent manner, which is indicative of a fully active preparation.

In our previously revised manuscript, we included a well validated proteinase K (PK) digestion assay to address the reviewer's concern regarding the quality of the sample preparation. We apologize that we didn't explain the PK digestion assay in more detail. We now incorporate this explanation and an updated schematic for this assay (Figure 2A below). MmCpn in the open state, i.e., in the absence of ATP, is known to be sensitive to PK digestion of the unstructured lid segments, yielding cleavage products (Lane 2). Upon ATP binding and hydrolysis, MmCpn closes the lid, which protects the segments from PK digestion. Release of ATP hydrolysis products, ADP and Pi, leads to reopening of the lid and restores some, but not all, PK sensitivity (Lane 3). **Therefore, the result that "there is still significant degradation" in the +ATP, as the reviewer commented, is exactly what is expected under cycling ATP conditions.** As we explained in the revision, this means that our preparation is fully active. This is the predicted behavior for a cycling chaperonin (doi: 10.1016/j.jmb.2015.04.013). Further, in lane 4, we carry out the PK digestion following incubation of MmCpn with ATP/AlFx which blocks cycling and locks the chaperonin in the closed state. Once MmCpn reaches the closed state under ATP

binding and hydrolysis, AIFx replaces the hydrolysis product Pi in the binding pocket, forming interactions with ADP and surrounding residues; this locks the MmCpn in the closed state and it becomes fully protected from proteinase K digestion. (doi: 10.1016/S0092-8674(00)81152-6). **Since we don't observe any digestion products in this condition, all the chaperonin molecules can reach the ATP-induced closed state. Therefore, we're confident that all particles in our preparation are functional and can be induced to a closed state upon ATP cycling.** Based on these three orthogonal assays, we are convinced and hopefully the reviewer will agree that our sample preparation is fully active and functional.

Figure 2. (A) Proteinase K digestion of incubations under three conditions. Lane 1 is a negative control. Lane 2 shows that MmCpn is digested by proteinase K in the open state in the absence of ATP. Lane 3 shows the PK digestion of MmCpn during ATP cycling. A small portion of MmCpn that exists transiently in the open state during ATP cycling can be digested by proteinase K, while the majority in the closed form is protected from proteinase K digestion. Lane 4 shows no observable proteinase K digestion, demonstrating that all particles in our preparation are functional and reach a closed state upon ATP binding and hydrolysis in the presence of AIFx. In the closed state, AIFx replaces ATP hydrolysis product Pi to interact with ADP and side chains and thus locks MmCpn in the closed state. (B) Rhodanese folding by MmCpn at 20 mins and 60 mins intervals. Data is normalized against the activity of 1 μM native rhodanese. Data are the mean ± s.e.m. of three independent replicates. (C) Nucleotide hydrolysis rates of MmCpn over a range of ATP concentrations. Rates shown were measured using an enzyme-coupled assay for the amount of ADP generated, which was calculated by monitoring NADH oxidation at 340 nm. Data shown as the mean ± 95% confidence intervals of three independent replicates. (D) Enlarged view of nucleotide hydrolysis rates of MmCpn at low ATP concentration range from Figure C.

The authors have not repeated experiments at variable ATP concentrations as recommended in the original reviews, leading to the weakening on their conclusion by the addition of the phrase "at the ATP condition that we study." (Line 108-109). They also chose not to use non-hydrolysable ATP analogs which would have been an additional assurance of functionality of the preparation by showing a fully closed condition in all of the images.

Response: We have answered these two questions in our first round of responses, additionally, our use of ATP/AlFx shows that our preparation is functional, see Figure 2 above.

REVIEWER COMMENTS

Reviewer #1 (Remarks to the Author):

This paper is the first, to the best of my knowledge, that shows that cryo-EM can be used to determine allosteric mechanisms by enabling the counting of the number of particles with different numbers of bound ligand molecules. This has been achieved before with native MS but cryo-EM also provides positional information. I'm, therefore, enthusiastic about publishing this work. The only issue that needs better clarification is that the results are seemingly in conflict with the work of J. Martin who found both intra- and inter-ring cooperativities for this system. The authors suggest that this may be due to binding vs. hydrolysis or the ATP concentration used in their experiments. The latter explanation cannot, however, account for the difference between the data in Fig. S1 and Martin's data in Fig. 5 (FEBS Lett. 2003). I think that a short paragraph on this discrepancy is needed in the Discussion.

Minor comment:

1. Line 131 – Redundant in what sense?

Reviewer #2 (Remarks to the Author):

The explanations given by the authors are very clear, and I'm happy with the, I don't have any further reservation.

Reviewer #3 (Remarks to the Author):

Although there have been efforts to address comments in the second round of reviews, significant flaws remain. Underlying these is the lack of response to the observation that more particles and thus higher resolution would improve confidence in the conclusion that ATP binding is stochastic. The relatively subtle conformational changes upon ATP binding are at the limit of observation in this study due to the poor resolution. The lack of resolution of the apical domains in Fig 1 and Fig S3 B is still of concern and the quality of the side views would be improved by processing more particles. Better resolution of the lid articulation would also allow conclusions about the role the ATP binding event on lid activation.

The doubts about quality control of the preparation in earlier reviews have now been addressed by the experiments and conclusions arising from Figure S1.

REVIEWERS' COMMENTS

Reviewer #1 (Remarks to the Author):

This paper is the first, to the best of my knowledge, that shows that cryo-EM can be used to determine allosteric mechanisms by enabling the counting of the number of particles with different numbers of bound ligand molecules. This has been achieved before with native MS but cryo-EM also provides positional information. I'm, therefore, enthusiastic about publishing this work. The only issue that needs better clarification is that the results are seemingly in conflict with the work of J. Martin who found both intra- and inter-ring cooperativities for this system. The authors suggest that this may be due to binding vs. hydrolysis or the ATP concentration used in their experiments. The latter explanation cannot, however, account for the difference between the data in Fig. S1 and Martin's data in Fig. 5 (FEBS Lett. 2003). I think that a short paragraph on this discrepancy is needed in the Discussion.

Response:

Thanks to the reviewer for bringing out this point. We agree that a discussion of discrepancy between previous publication and our study is necessary to clarify the manuscript. We now highlight such discussion in the manuscript.

In Supplementary Figure 1C, we determined the steady state ATPase activity of MmCpn with NADH-coupled enzymatic assay according to a previously published protocol (Lopez et al Nature SMB, doi:10.1038/nsmb.3440) to demonstrate our sample is indeed biochemically active. Our data agrees with the previous data (Figure 3 in Lopez *et al*/ Nature SMB, Figure 4 in Reissmann *et al*/ Nature SMB, doi:10.1038/nsmb1236). In Martin's study (FEBS Lett. 2003), the ATPase activity was measured differently. All these results showed the activity assay plots deviated a bit as the reviewer pointed out. Nevertheless, the conclusions of all the previous papers and our assay supported nested cooperativity, that is, intra-ring positive and inter-ring negative. Therefore, there is no conceptual disagreement in the conclusion between our biochemical data (Supplementary Figure 1) and Martin's, and is mentioned in the revised text under Results.

As the ATPase assay conflates the ATP binding and hydrolysis, the cooperativity observed also conflates both steps. To examine whether MmCpn has such nested cooperativity in ATP binding (and not its ATPase activity), we purposefully performed our study at an intermediate ATP saturation level to achieve ATP occupancy in individual particles ranging from 0 to 100% (Figure 5a and b). Our study demonstrates that ATP binding in MmCpn is stochastic both within and across the rings for the full spectrum of particle ATP saturation at the ATP concentration studied. To perform the ATPase activity assay (Supplementary Figure 1C and Martin's data) to demonstrate the functionality of our sample, we could not use the ATP concentration we used in our cryoEM study. Our image data analysis suggests that if the ATP concentration was higher, even by a factor of two, the ATP sites would be saturated on virtually all particles. Therefore, such a condition would not yield information to our question of investigation whether binding

was cooperative or not. Once all the sites are fully occupied with ATP, hydrolysis could very well be cooperative, as our data and Martin's paper suggest.

Minor comment:

1. Line 131 – Redundant in what sense?

Response:

Thank you for the question, we will clarify in the manuscript with the following explanation:

To avoid the overrepresentation of the images with two major preferred orientations in the final reconstructions, it is customary to equalize the contributions of different particles' orientations by reducing the number of included particles with these orientations for the final reconstruction.

Reviewer #2 (Remarks to the Author):

The explanations given by the authors are very clear, and I'm happy with the, I don't have any further reservation.

Response: Thank you.

Reviewer #3 (Remarks to the Author):

Although there have been efforts to address comments in the second round of reviews, significant flaws remain. Underlying these is the lack of response to the observation that more particles and thus higher resolution would improve confidence in the conclusion that ATP binding is stochastic.

Response: The theme of our manuscript is to determine which and how many of the subunits in each individual particle contains ATP. ATP binding is a binary observation. Our analysis is based on ~630,000 MmCpn subunits with ATP-bound and ~300,000 without. We believe that our analysis allows us to draw statistically valid conclusions. This is supported by the clear presence of ATP density in the four "closed" conformational classes and the absence of ATP density in the other four "open" conformational classes.

The relatively subtle conformational changes upon ATP binding are at the limit of observation in this study due to the poor resolution.

Response: We do not believe that the observed conformational differences in the 8 classes are subtle since the open and close states of the apical domain vary up to 35 Å (Figure 2c). Interestingly, the conformational difference appears to make intuitive sense even in our moderate resolution data because it has been shown that the lid (apical domain) of the subunit can undergo from open to closed conformation upon ATP binding. Therefore, we disagree that significant flaws remain in our revised manuscript. We are pleased that the other two reviewers did not object to our responses regarding the number of particles in the last round of our manuscript revision.

The lack of resolution of the apical domains in Fig 1 and Fig S3 B is still of concern and the quality of the side views would be improved by processing more particles. Better resolution of

the lid articulation would also allow conclusions about the role the ATP binding event on lid activation.

Response: As we pointed out in the previous two responses, ATP binding induces large heterogeneity in the upper domain of MmCpn particles, both in individual particles and among particles. The reconstructions shown in Figure 1 and Supplementary Figure 3b average large number of such heterogeneous particles, and as expected, has the upper domain including the apical domain poorly resolved ($>10 \text{ \AA}$). When classified into individual subunits (Figure 2b) rather than the entire ring (Figure 1, Supplementary Figure 3), the resolution is improved to 4-6 \AA , therefore, there is sufficient resolution in the apical domains when treated as individual subunits, which is impossible to do for the ring as a whole. Therefore, the lack of resolution in the apical domain in the reconstructions with both rings or single ring is the result of heterogeneity, and the resolvability of this region in the averaged reconstructions shown in Fig 1 and Supplementary Fig 3B would not be improved by averaging more particles.

The doubts about quality control of the preparation in earlier reviews have now been addressed by the experiments and conclusions arising from Figure S1.

Response: Thank this reviewer for accepting our responses on the biochemical analysis of protein activity validation.